# Genomic and epigenomic integrative subtypes of renal cell carcinoma in a Japanese cohort

Akihiko Fukagawa[1,2], Natsuko Hama[1], Yasushi Totoki [1,3], Hiromi Nakamura[1], Yasuhito Arai [1], Mihoko Saito-Adachi [1], Akiko Maeshima[4], Yoshiyuki Matsui[5], Shinichi Yachida [3], Tetsuo Ushiku [2] & Tatsuhiro Shibata [1,6] ✉

Renal cell carcinoma (RCC) comprises several histological types characterised by different genomic and epigenomic aberrations; however, the molecular pathogenesis of each type still requires further exploration. We perform whole-genome sequencing of 128 Japanese RCC cases of different histology to elucidate the significant somatic alterations and mutagenesis processes. We also perform transcriptomic and epigenomic sequencing to identify distinguishing features, including assay for transposase-accessible chromatin sequencing (ATAC-seq) and methyl sequencing. Genomic analysis reveals that the mutational signature differs among the histological types, suggesting that different carcinogenic factors drive each histology. From the ATAC-seq results, master transcription factors <u>are</u> identified for each histology. Furthermore, clear cell RCC <u>is</u> classified into three epi-subtypes, one of which expresses highly immune checkpoint molecules with frequent loss of chromosome 14q. These genomic and epigenomic features may lead to the development of effective therapeutic strategies for RCC.

Renal cell carcinoma (RCC) affects nearly 330,000 people worldwide yearly, with more than 140,000 patients with RCC estimated to die each year. Different histological types of RCC are associated with different genetic alterations, epigenomic aberrations, clinical outcomes, and therapeutic responses[1,2]. The recognised environmental and lifestyle risk factors for RCC include obesity, smoking, hypertension, and diabetes[3]. Additionally, aristolochic acid, an abundant compound in *Aristolochia* plants and natural herbs, reportedly contributes to mutational processes in Romanian, Chinese, and Japanese RCC cases[4,5].

RCC is characterised by fewer single nucleotide variants (SNVs) than other types of cancer, with more frequent short insertions/deletions (indels) and somatic copy number alterations (SCNAs)[6]. Clear cell RCC (ccRCC) is the most common type of RCC (~75%). It shows a significant loss of chromosomes (chr.) 3p, encoding *VHL*, in addition to

the gain of chr.5q[7]. Papillary RCC (PRCC) is the second most common type (~20%) and is characterised by chromosomal amplifications such as gains of chr.7 and chr.17 and chromosomal loss of chr.22q[8]. Chromophobe RCC (ChRCC) arises from the distal nephron. It is a relatively rare type (~5%) characterised by widespread chromosomal losses, including chr.1, chr.2, chr.6, chr.10, chr.13, and chr.17, mitochondrial DNA alterations, and reliance on oxidative phosphorylation[9]. MiT family translocation-positive RCC is a rare tumour that harbours gene fusions involving *TFE3* or *TFEB* but accounts for approximately 40% of paediatric RCC and less than 5% of adult RCC[10,11].

Epigenomic aberrations in RCC have attracted increased attention recently[12–14]. In particular, chr.3p, whose loss is a typical genomic aberration in ccRCC, contains several chromatin modifiers, such as PBRM1, SETD2, and BAP1, and these genetic mutations are also

[1]Division of Cancer Genomics, National Cancer Center Research Institute, Tokyo, Japan. [2]Department of Pathology, Graduate School of Medicine, The University of Tokyo, Tokyo, Japan. [3]Department of Cancer Genome Informatics, Graduate School of Medicine, Osaka University, Osaka, Japan. [4]Department of Diagnostic Pathology, National Cancer Center Hospital, Tokyo, Japan. [5]Department of Urology, National Cancer Center Hospital, Tokyo, Japan. [6]Laboratory of Molecular Medicine, The Institute of Medical Science, The University of Tokyo, Tokyo, Japan. ✉e-mail: tshibata2010@gmail.com

frequently observed in the contralateral allele[15,16]. Therefore, further epigenomic studies are required to improve our understanding of RCC.

Recently, inhibitors of immune checkpoint molecules such as programmed death-1 (PD-1) have been widely adopted to treat various malignant tumours. Tumour mutation burden, neoantigen load, and degree of CD8 + T-cell infiltration have been suggested as effective biomarkers for predicting clinical response[17–19]. However, these conventional biomarkers have not been associated with the clinical response in RCC[20,21]. Other genetic alterations have been suggested as potential biomarkers for predicting drug responses; however, they are not yet thoroughly understood[21–23].

Here, we report the whole-genome profiling of 128 Japanese RCC cases and transcriptomic profiling of more than 200 cases, including ccRCC, PRCC, ChRCC, and TFE3-translocated RCC (TFE3-RCC). Furthermore, we perform epigenomic sequencing analyses using an assay for transposase-accessible chromatin sequencing (ATAC-seq) and enzymatic methyl sequencing (EM-seq)[24]. This study clarifies the mutagenesis processes, genomic alterations, and epigenomic profiles characteristic of each histological type. We also identify three ccRCC epi-subtypes associated with a unique immune environment. These molecular characteristics may lead to effective diagnostic, therapeutic, and preventive strategies for RCC treatment.

## Results

### Somatic genomic aberrations in Japanese RCCs
We conducted transcriptomic sequencing (RNA sequencing; RNA-seq) of 287 Japanese RCC cases, including 258 ccRCC, 17 PRCC, and 12 ChRCC cases (Supplementary Data 1). We also performed whole-genome sequencing (WGS) analysis of 128 Japanese RCC cases (102 ccRCC, 15 PRCC, 11 ChRCC) (Supplementary Data 2). Consequently, we detected TFE3 fusions using RNA-seq in three PRCC cases and identified the histological type as TFE3-RCC (Supplementary Data 3 and 4). A total of 857,020 SNVs and 44,656 indels were identified using WGS.

The global driver landscape of somatic alterations in this cohort is shown in Fig. 1a. Consistent with previous reports[4,7,15,16,25], our study confirmed frequent alterations in key ccRCC drivers, including VHL (71%), PBRM1 (38%), BAP1 (17%), SETD2 (14%), KDM5C (8%), and TP53 (6%) mutations (Fig. 1a, Supplementary Fig. 1a, and Supplementary Data 7). Moreover, TOPORS mutations were observed in four ccRCC cases (Fig. 1a), and all TOPORS mutated cases had necrosis, a poor prognostic factor for ccRCC[26]. Ninety percent of ccRCC cases exhibited a loss of chr.3p, a gain of chr.5q (61%), a loss of chr.14q (39%), and a gain of chr.7 (35%) were also frequent (Fig. 1a and Supplementary Fig. 1b). By contrast, ChRCC showed fewer SNVs and indels. Still, as in previous reports[7,9], frequent TP53 mutations (36%) and losses of chr.1p (90%), chr.2 (100%), chr.6q (100%), chr.10 (80%), chr.13 (60%), chr.17 (100%), and chr.21 (70%) (Fig. 1a, Supplementary Fig. 1c, and Supplementary Data 9). PRCC had no frequent nonsynonymous mutations but showed losses of chr.14q (46%), chr.18q (46%), and chr.22q (46%) and gain of chr.16 (46%) and chr.17 (46%) (Fig. 1a).

Next, we investigated genomic aberrations associated with tumour progression in ccRCC. Stages I−II and III−IV were defined as early-stage and advanced-stage cases, respectively. Although the number of genome-wide SNVs and indels was not significantly different between the two groups (Supplementary Fig. 1d), ZFHX3, SETD2, and TP53 mutations were observed more frequently in advanced-stage cases (P = 1.03E-02, odds ratio (OR) = 11.5, P = 2.69E-02, OR = 3.7, and P = 2.1E-03, OR=inf, respectively) (Fig. 1b and Supplementary Data 11). In advanced ccRCC cases, the number of SCNAs across the genome increased (P = 1.30E-06), in particular losses of chr.1p (P < 0.05), chr.4 (P < 0.05), chr.9 (P < 0.05), chr.13 (P < 0.001) and chr.14q (P < 0.01) were observed more frequently (Fig. 1c, Supplementary Fig. 1e, and Supplementary Data 13).

### Mutational signature analysis of Japanese RCCs
Associations between histological type, lifestyle, and genetic polymorphisms were investigated (Table 1, top). Patients aged at diagnosis (over 60 years old) were more likely to have ccRCC (P = 5.63E-03), and ccRCC was more frequent in patients diagnosed with diabetes (P = 2.57E-02). Furthermore, as moderate alcohol consumption reportedly reduces the risk of RCC[27,28], we examined the association between genes involved in alcohol metabolism. ccRCC and PRCC were more frequent in patients with the single nucleotide polymorphism (SNP) rs1229984 in subunit 1 B of the alcohol dehydrogenase gene (ADH1B), which leads to the inactivation of alcohol metabolism[29] (P = 1.87E-03). In contrast, ChRCC was frequent in patients at a relatively young age at diagnosis (P = 2.20E-03) and in those with aldehyde dehydrogenase 2 (ALDH2) dysfunction (rs671) (P = 1.11E-02).

To clarify the mutational processes in each histological type, we performed mutational signature analysis using FitMS with GEL-kidney-SBS in the organ-specific signature[30]. Sixteen single-base substitutions (SBS) signatures were identified (Fig. 2a). In ccRCC and PRCC cases, SBS5 (clock-like signature) and SBS125 (unknown cause) were predominant (P = 8.2E-04 and P = 8.5E-04, respectively). Both signatures were positively correlated with age at diagnosis (P = 2.6E-04 and P = 2.78E-02, respectively) (Table 1, bottom). SBS125 was also correlated with ADH1B genotype (rs1229984) (P = 2.44E-02). Furthermore, SBS107 (a signature similar to tobacco exposure-associated SBS4) significantly contributed to ccRCC (P = 3.0E-04). However, we did not observe a correlation with smoking. In contrast, SBS17 was highly enriched in ChRCC (P = 4.34E-02).

Previous studies have reported that aristolochic acid exposure associated with SBS22 is a risk factor for RCC, and several Japanese ccRCC cases have demonstrated SBS22[4,5]. SBS22 was detected in one case in this cohort. Patients with bi-allelic ADH1B genotype (rs1229984) frequently exhibited SBS117 (unknown cause) (P = 5.46E-03); however, this signature did not correlate with alcohol consumption. Other epidemiological risk factors, including hypertension, obesity, and diabetes, were not associated with specific mutational signatures.

To validate our results, we analysed the 186 samples from The Pan-Cancer Analysis of Whole Genomes (PCAWG) study[31] (Supplementary Fig. 2a and Supplementary Data 14). Similar to our cohort, SBS107 and SBS125 were predominant in ccRCC (P = 3.17E-15 and P = 1.12E-02, respectively), and SBS125 contributed highly to PRCC compared to ccRCC (P = 3.26E-08), but SBS107 contributed little (P = 1.16E-12). In addition, the contribution of SBS125 was limited in ChRCC (P = 1.91E-18).

### Chromatin status-based classification of RCC
We examined 20,000 regions with significant variance between −100,000 and +1000 base pairs (bp) from the transcription start site (TSS) from ATAC-seq data. We performed hierarchical clustering for 66 Japanese cases (Supplementary Data 15) and 50 cases deposited in The Cancer Genome Atlas (TCGA) using cancerPeaks[32] to identify the features of chromatin status in each histological type (Fig. 3a and Supplementary Data 16 and 17). Six clusters (peak clusters 1−6) of chromatic status were identified with unique enrichments of characteristic transcription factor binding motifs and nearby genes associated with the histological subtypes. In 64 cases that underwent ATAC-seq analysis, we extracted recurrent hypomethylated regions (HMRs) from methyl sequencing data and performed hierarchical clustering using the methylation rate. Consequently, six clusters (methyl-clusters 1−6) of methylation status were identified (Supplementary Fig. 4a and Supplementary Data 18).

### Molecular features of three ccRCC epi-subtypes
Using ATAC-seq analysis, ccRCC was classified into three epi-subtypes (cc_1-3) based on the combination of peak clusters 2 and 3: cc_1 showed moderate intensities of both peak clusters 2 and 3, cc_2 showed the

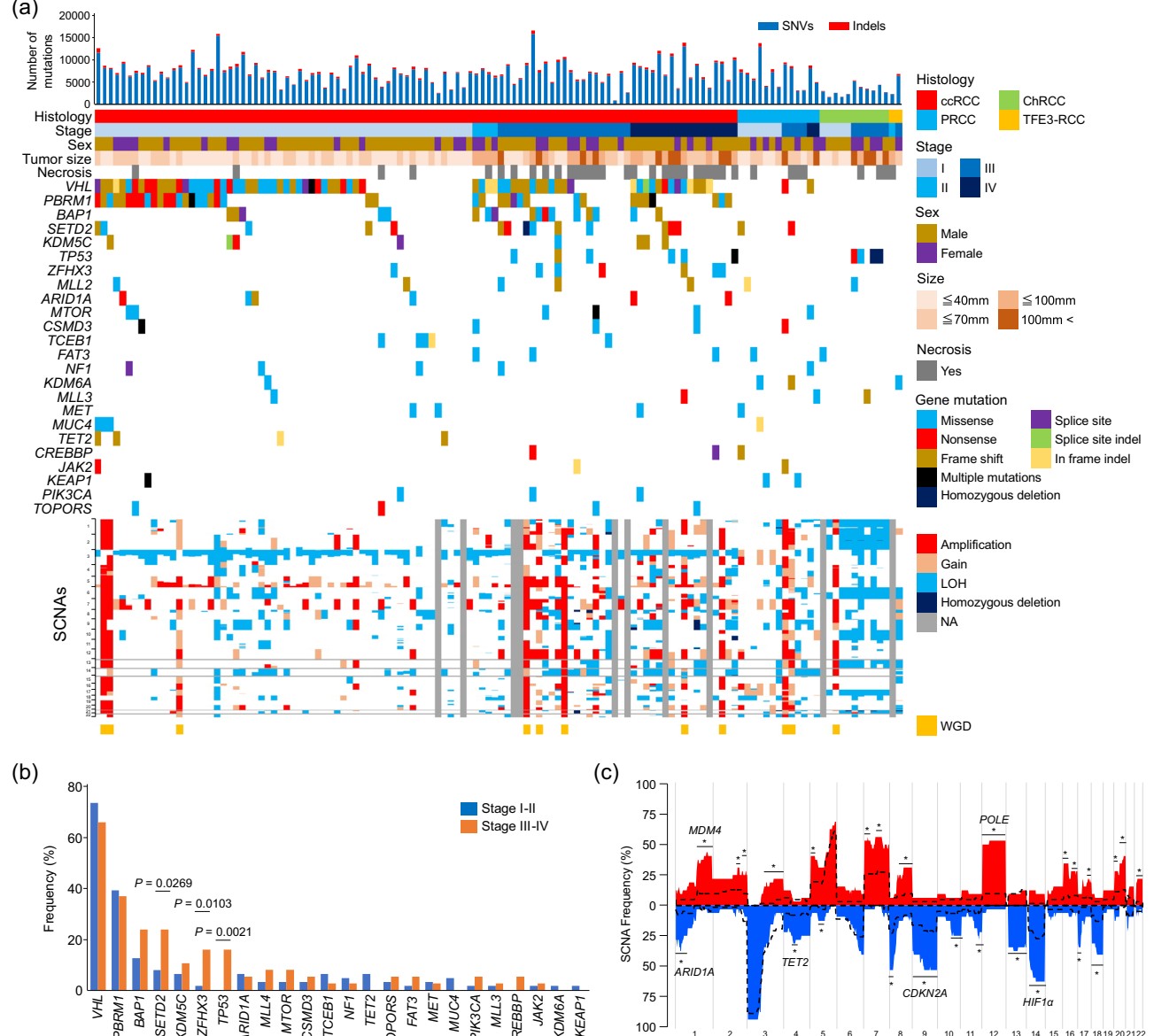

**Fig. 1 | Landscape of somatic alterations in renal cell carcinoma (RCC) cases. a** Somatic alteration landscape of 128 Japanese RCC cases. The top histogram represents the number of single nucleotide variants (SNVs) and short indels across the whole genome. The top heatmap's upper part shows histology, sex, stage, tumour size, and necrosis. The lower part of the top heatmap shows frequent nonsynonymous mutations. The bottom heatmap shows the somatic copy number alterations (SCNAs). N/A, not applicable; LOH, loss of heterozygosity; WGD, whole genome doubling. **b** Frequencies of advanced-stage and early-stage cases with nonsynonymous mutations. *P*-values and odds ratios (ORs) were calculated using the two-sided Fisher's exact test. **c** Frequency of ccRCC cases with SCNAs. Red and blue indicate the somatic copy number gains and deletions, respectively, in advanced-stage cases. The dotted lines indicate those of early-stage cases. *P*-values were calculated per cytoband using the two-sided Fisher's exact test. The areas where significant differences (*P*-value < 0.05) were observed in multiple cytobands consecutively were marked by an asterisk (*). *P*-values were shown in Supplementary Data 13. Source data are provided as a Source Data file.

high intensity of peak cluster 2, and cc_3 had high peak cluster 3 intensity.

Footprint analysis using TOBIAS[33] also indicated that immune-related genes such as the interferon-regulatory factor (IRF) family and signal transducer and activator of transcription (STAT) 1 and 2 were pivotal molecules in cc_2 (Supplementary Fig. 5a, b). Moreover, hepatocyte nuclear factor 1 homeobox A (HNF1A) and B (HNF1B) were more abundant in cc_3 than in cc_1 (Supplementary Fig. 6a, b). In contrast, cc_3 was characterised by the enrichment of the *HIF* gene family and its downstream target genes, including *VEGFA*, which are involved in the hypoxic response (Supplementary Fig. 3b).

While there were no significant differences in the number of SNVs and indels among the three epi-subtypes (Supplementary Fig. 7a), we

detected significant differences in several SCNAs and nonsynonymous mutations. From the result of multivariate analysis, no specific nonsynonymous mutations were associated with cc_2 (Supplementary Data 19). However, deletions of chr.14q, containing *HIF1α*, and gain of chr.7q (between 7q31.31 and 7q36.3) were more and less frequent in cc_2 than in cc_1 and cc_3, respectively (*P* = 5.10E-03 and *P* = 4.38E-02, respectively) (Fig. 4a and Supplementary Data 19 and 20). cc_3 was characterised by SWI/SNF family gene mutations and gain of chr.7q (between 7q31.31 and 7q36.3), containing *MET* (*P* = 5.72E-03 and *P* = 1.63E-02, respectively). cc_1 featured a smaller tumour size than the other subtypes, and cc_2 and cc_3 featured relatively low expression of chromatin modifiers and tumour suppressor genes compared to cc_1 (Supplementary Figs. 7b, 8a, and 8b). In addition, gene set enrichment

**Table 1 | Multivariate analyses of histology and substitution mutational signature**

| Histology | Field | Category | P-value |
|---|---|---|---|
| ccRCC | Diagnosis age | 60< | 5.63E-03 |
| ccRCC | Diabetes | Yes | 2.57E-02 |
| ChRCC | Diagnosis age | low | 2.20E-03 |
| ChRCC | ALDH2 | ALT/ALT | 1.11E-02 |
| not ChRCC | ADH1B | REF/ALT or ALT/ALT | 1.87E-03 |
| Signature | Field | Category | P-value |
| SBS1 | Diagnosis age | 70< | 2.52E-02 |
| SBS5 | Histology | not ChRCC | 8.2E-04 |
| SBS5 | Diagnosis age | 60< | 2.6E-04 |
| SBS17 | Histology | ChRCC | 4.33E-02 |
| SBS17 | Diagnosis age | ≤40 | 2.02E-02 |
| SBS107 | Histology | ccRCC | 3.0E-04 |
| SBS107 | Tumour size | 70 mm< | 1.83E-02 |
| SBS117 | ADH1B | ALT/ALT | 5.46E-03 |
| SBS117 | Diagnosis age | 60< | 1.66E-02 |
| SBS125 | Histology | not ChRCC | 8.5E-04 |
| SBS125 | Diagnosis age | High | 2.78E-02 |
| SBS125 | ADH1B | REF/ALT or ALT/ALT | 2.44E-02 |

Top, multivariate analysis of histology; bottom, that of mutational signature. Factors that showed significant differences ($P < 0.05$) are listed. $P$-values were not adjusted.

analysis (GSEA) comparing cc_1 to the other subtypes showed that intercellular and stromal adhesions were downregulated in cc_2 and cc_3 (Supplementary Fig. 8c).

Clinically, a significantly higher rate of early recurrence was observed in cc_2 patients (hazard ratio (HR) = 9.26, 95% confidence interval (CI) = 1.137−75.44, $P$ = 1.1E-02) (Fig. 4b). Since immune-related genes, such as *IRF* family and C-X-C Motif Chemokine Receptor 4 (*CXCR4*), were enriched in cc_2 (Fig. 3a and Supplementary Fig. 5a), we performed CIBERSORTx to assess the immune environment of each epi-subtype. Immune cell infiltration was not significantly different among the three epi-subtypes, but immune fractions such as T follicular helper cells (Tfh) and regulatory T cells (Treg) were increased in cc_2 (Fig. 4c, Supplementary Fig. 9a, b). Furthermore, the expression levels of *PD-1*, cytotoxic T-lymphocyte-associated protein 4 (*CTLA4*), and lymphocyte activation gene 3 (*LAG3*) were markedly higher in cc_2 (Fig. 4d). In comparison, there was no significant difference in the expression of programmed death-ligand 1 (*PD-L1*) (Supplementary Fig. 9c). Immunohistochemistry (IHC) verified an increase in PD-1-positive lymphocytes among the tumour-infiltrating inflammatory cells in cc_2 (Fig. 4e and Supplementary Fig. 9d).

## Chromatic features identified master transcription factors in PRCC, ChRCC, and TFE3-RCC

PRCC was characterised by the enrichment of peak clusters 4−6, which showed strong enrichment of AP-1 (FOS/JUN) and TEAD activities (Figs. 3a and 5a). Footprint analysis verified that the binding motifs of the FOS and JUN gene families were present in PRCC cases. Still, no significant differences were observed in the TEAD gene family (Supplementary Fig. 10a). Hierarchical clustering analysis showed that the two TFE3-RCC cases had chromatic characteristics similar to those of the PRCC. Previous studies have shown that TFE3 fusion protein is associated with tumour progression by promoting its nuclear translocation[11,34,35]. Therefore, we calculated the binding score of the TFE3 motif using TOBIAS[33] and determined that TFE3 had a higher DNA-binding activity in TFE3-RCC than in PRCC (Fig. 5b, c).

ChRCC cases were enriched in peak cluster 1, which showed characteristic enrichment of the binding motifs of the forkhead box

(FOX) gene family, grainyhead-like transcription factor 2 (GRHL2), and oestrogen-related receptor gamma (ERRγ) (Fig. 3a and Supplementary Fig. 3a). A comparison of the binding scores between ChRCC and ccRCC confirmed the higher activities of these transcriptional factors (Fig. 6a, b, and Supplementary Fig. 11a). Whole-genome methylation sequencing analysis also revealed that CpG sites surrounding the binding motifs of the FOX gene family, GRHL2, and ERRγ were significantly hypomethylated in ChRCC (Supplementary Fig. 4a).

GRHL2 is highly expressed in the surface ectoderm and adult renal collecting duct cells. It regulates the epithelial barrier function of the collecting duct and renal osmoregulation[36−38]. *GRHL2* was highly expressed in ChRCC, and its promoter region was hypomethylated (Fig. 6c, d). The expression of protocadherin 1 (*PCDH1*) and serine peptidase inhibitor, Kunitz type 1 (*SPINT1*), both of which are verified target genes of GRHL2[39], was also significantly higher in ChRCC than in other RCCs (Fig. 6e). *GRHL2* expression promotes chromatin accessibility in the gene bodies of *PCDH1* and *SPINT1*[39]. These regions consistently showed increased chromatin accessibility in ChRCC (Supplementary Fig. 11d).

ERRγ interacts with peroxisome proliferator-activated receptor γ coactivator-1 (PGC-1), peroxisome proliferator-activated receptor (PPAR), and FOXO1 to enhance mitochondrial oxidative phosphorylation and fatty acid metabolism[40,41]. PCG-1/ERR signalling is involved in the metabolic reprogramming of malignant tumours[41]. *ERRγ* was highly expressed, and its promoter region was hypomethylated in ChRCC (Fig. 6f, g), and *PGC-1α*, *PPARγ*, and *FOXO1*, all of which are co-activators of ERRγ, were also highly expressed in ChRCC (Supplementary Fig. 11e). GSEA showed that the genes involved in oxidative phosphorylation, fatty acid metabolism, and tricarboxylic acid cycle pathways were more highly expressed in ChRCC than in ccRCC (Fig. 6h). These results suggest that *GRHL2* and *ERRγ* contribute to the unique cell adhesion, prominent cell border in histopathology, and a unique metabolism in ChRCC.

## Identification of promoter mutations that increase chromatin accessibility

We conducted an integrative analysis of ATAC-seq and WGS data for our samples to identify promoter mutations that affect chromatin accessibility. We attempted to identify accessibility-enhancing mutations based on the differences in variant allele frequencies (VAFs) of the mutations in ATAC-seq and WGS and the degree of chromatin accessibility (Supplementary Fig. 12a)[32,42]. Of the 857,020 SNVs detected by WGS, 1,078 had ten or more reads in ATAC-seq and were located between −2000 bp and +100 bp from the TSS. No recurrent promoter SNVs increased chromatin accessibility (Fig. 7a). We found a *TERT* promoter mutation in one ccRCC sample (sample ID: KI253) whose *TERT* gene expression level was the second highest in our cohort (Fig. 7b). This mutation has not been a major one associated with high *TERT* expression in other tumour types[43,44]. We further examined the differential motif scores between the wild-type and mutated *TERT* promoter regions to explore potential transcriptional factors whose binding affinity was significantly altered (Supplementary Fig. 12b). The motif scores of the ETS and HIF gene families were higher in mutated promoters than in the wild-type promoters (Fig. 7c, d and Supplementary Data 21), suggesting that this *TERT* mutation may enhance chromatin accessibility and gene expression by acquiring higher affinity with HIF and ETS gene families.

Extending this differential motif score analysis, we attempted to identify transcriptional factors with higher binding affinities for promoter mutations in ccRCC. We defined the promoter mutations, which showed higher ATAC-seq VAF to WGS VAF and higher accessibility, as accessibility-enhancing SNVs (aeSNVs). The expression of 84 genes with aeSNVs was significantly higher than that of genes with non-aeSNVs ($P$ = 67.34E-05) (Fig. 7e). By comparing the binding affinities between the aeSNVs and wild-type in each case, we found that

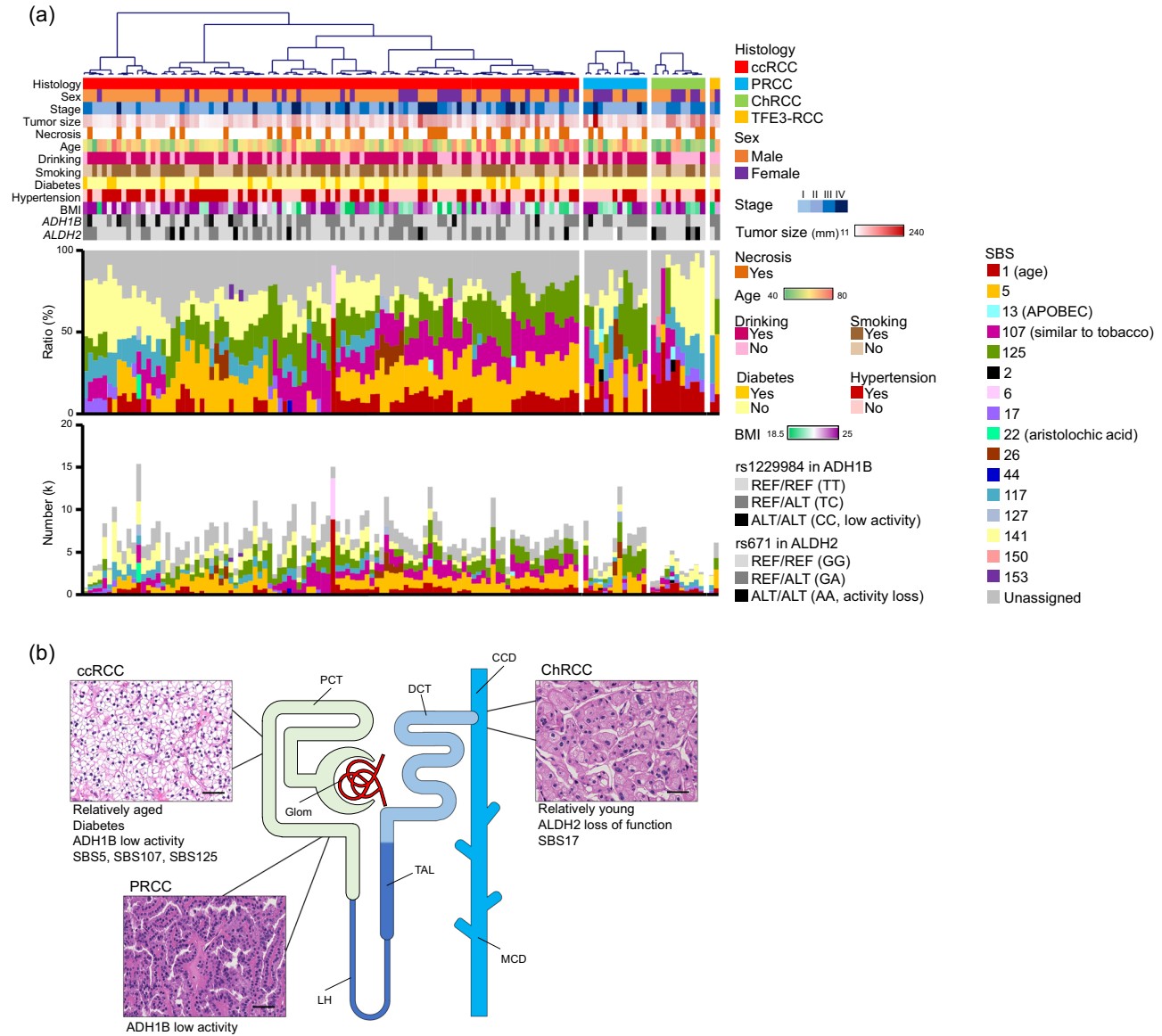

**Fig. 2 | Substitution mutational signatures in Japanese RCC cases. a** Signal-mutational signatures of 128 Japanese RCC cases. The heatmap represents clinical information and single nucleotide polymorphisms (SNPs) in *ALDH2* and *ADH1B*. REF and ALT represent reference and alternative alleles, respectively. BMI, mass index. The upper and lower histograms show the relative and absolute values of the mutational signatures, respectively. SBS, single-base substitutions. **b** A cartoon of nephron and features of RCC subtypes. Glom glomerulus, PCT proximal convoluted tubule, LH loop of Henle, TAL thick ascending limb of Henle's loop, DCT distal convoluted tubule, CCD cortical collecting duct, MCD medullary collecting duct. Scale bars in HE, 50 μm. Source data are provided as a Source Data file.

the motif score of ZEB2, a key regulator of epithelial-mesenchymal transition[45], increased in 17.8% of cases with aeSNVs. This rate was significantly higher than that for non-aeSNVs (7.8%) ($P = 4.00\text{E-}03$, OR = 2.6) (Fig. 7f). Significantly increased frequencies of higher binding affinities to aeSNVs were also observed for HIF2α, ZEB1, and HIF1β ($P = 3.83\text{E-}02$, OR = 2.3, $P = 3.89\text{E-}02$, OR = 2.2, and $P = 4.97\text{E-}02$, OR = 2.9, respectively). HIF1α and HIF2α are accumulated by the bi-allelic inactivation of *VHL*. HIF2α has been implicated as a promoter of aggressive tumour behaviours by regulating genes associated with lipoprotein metabolism, ribosome biogenesis, and MYC transcriptional activities, while HIF1α is a suppressor[46,47]. ZEB2, HIF2α, and ZEB1 motifs were also significantly enriched in aeSNVs using the permutation test (Fig. 7g). Our results suggest that ZEB2 and HIF2α may also be involved in the global regulation of promoter-mutated genes in ccRCC.

Furthermore, we performed GSEA to elucidate the biological function of genes with aeSNVs, which were enriched in nuclear transport-related genes (Supplementary Fig. 12c). In contrast, genes with non-aeSNVs were enriched in cell adhesion genes.

## Discussion

A comprehensive understanding of RCC histological types' common and distinct features is critical for patient management and therapeutic strategies. In this study, we performed integrated genomic, transcriptomic, and epigenomic analyses of various histological types. We demonstrated that the morphological classification of RCC reflects differences in genomic alterations, mutational processes, and chromatin status.

The prominent variants in Japanese ccRCC were *VHL*, *PBRM1*, *BAP1*, and *SETD2*, as well as chr.3p loss and chr.5q gain, similar to previous large studies, including the TCGA study[7,15,16,25,48]. Advanced-stage ccRCC cases had more SCNAs, although the number of SNVs and indels did not show a statistically significant difference, suggesting that chromosomal instability (CIN) contributes more strongly to

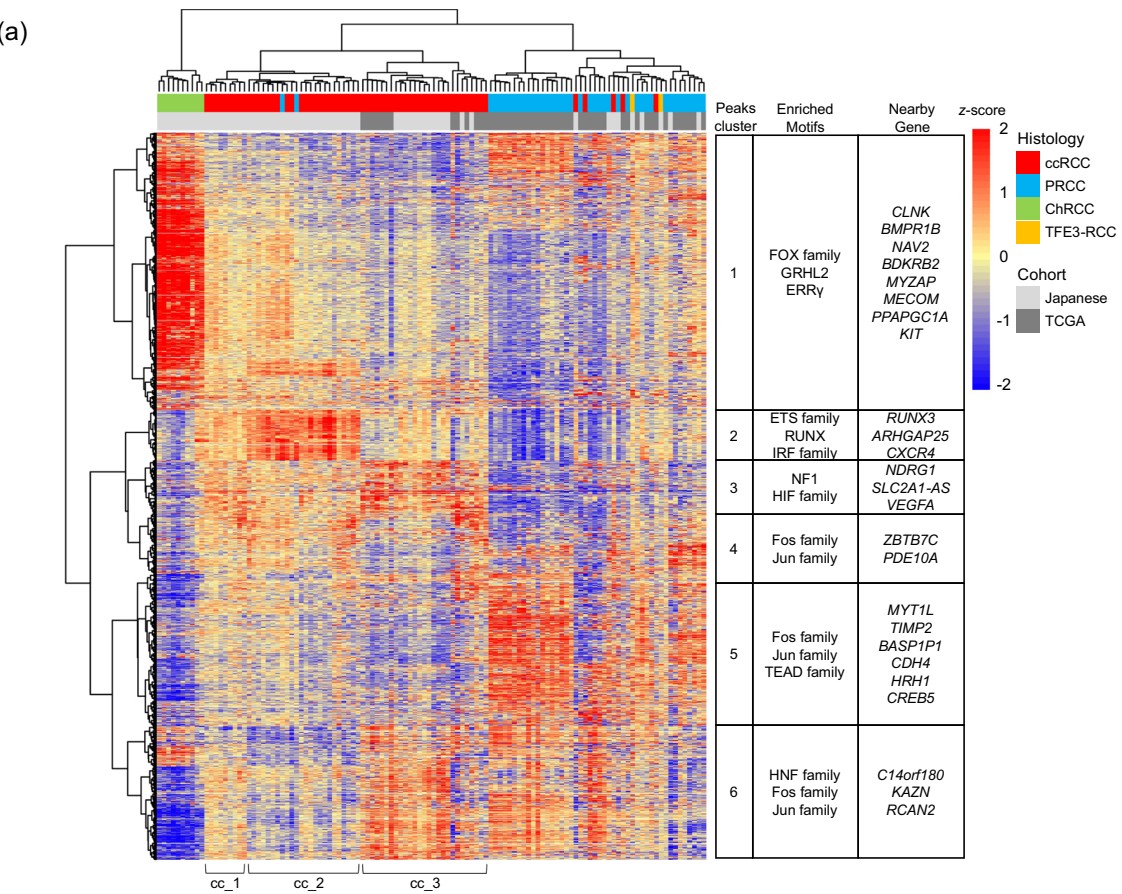

**Fig. 3 | Hierarchical clustering of assay for transposase-accessible chromatin sequencing (ATAC-seq) data. a** Heatmap of ATAC-seq accessibility across all RCC samples in this study and data downloaded from The Cancer Genome Atlas. Each row represents a 501-bp region registered in cancerPeaks, and each column represents a case. Heatmap colours indicate the *z*-scores of peak intensities for each sample. Hierarchical clustering was performed using the ward.D2 method and Euclidean distance. Cohort indicates the sources of samples. Twenty thousand peaks are divided into six peak clusters, and enriched motifs and nearby genes in each peak cluster are shown on the right. Source data are provided as a Source Data file.

tumour progression. *ZFHX3*, *SETD2*, and *TP53* mutations were frequently found in advanced-stage ccRCC. Loss of SETD2 and p53 function correlates with aggressive clinicopathological features and poor overall survival[49–52], and these mutations induce CIN in ccRCC[53,54]. ATBF1, encoded by *ZFHX3*, negatively regulates *AFP* and *MYB*, positively regulates *CDKN1A* expression, and has been reported as a tumour suppressor gene in other types of cancer[55,56]. Therefore, the loss of function of this molecule may also promote tumour progression in ccRCC, and further studies are needed to elucidate this mechanism. *TOPORS* mutated cases had more necrosis, a poor prognosis factor for ccRCC, and this mutation may also promote CIN because the gene regulates homologous recombination repair[57].

Although PRCC has been divided into morphological types 1 and 2 for approximately two decades[58], this classification has yet to be recommended recently because type 2 PRCC consists of individual subgroups with different molecular backgrounds[2,8]. In this study, we were unable to identify frequent nonsynonymous mutations. We frequently detected SCNAs in several chromosomal regions commonly observed in ccRCC, especially in advanced-stage cases, and could not be determined as characteristics of PRCC. ChRCC cases showed genome-wide LOH; one showed whole-genome doubling (WGD). The major copy numbers of chr.2, chr.6, chr.10, and chr.17, in the whole genome doubled case, are neutral, and it is presumed that LOH occurred before the WGD, suggesting that these LOH may be an early event in ChRCC.

Mutational signature analysis for our cohort and PCAWG samples showed that mutagenesis processes differed among histological types.

SBS117 and SBS141 contributed highly in Japanese samples and were not detected in PCAWG cases, suggesting that these are characteristic of the Japanese cases. ChRCC showed a distinctly different mutational signature from other histological types, which is consistent with the fact that the distal nephron where ChRCC arises is predicted to be exposed to varying concentrations of potentially carcinogenic substances than the proximal nephron where ccRCC and PRCC occur (Fig. 2b). ChRCC was observed more frequently in patients with ALDH2 dysfunction. These patients did not have drinking habits. ALDH2 expression in ChRCC was relatively low compared to other subtypes (Supplementary Fig. 11c). ALDH2 also metabolises endogenous aldehydes such as 4-hydroxy-2-nominal and malondialdehyde produced by lipid peroxidation, and these products are endogenous mutagens that cause *TP53* mutations and CIN[59,60]. Therefore, these aldehydes, including acetaldehyde, may influence ChRCC development.

In contrast, low ADH1B activity would affect ccRCC and PRCC arising from the proximal nephron. Large-scale epidemiological studies have shown that diabetes is a risk factor for RCC in other ethnicities[61,62], and diabetes may be associated with an increased risk for the development of ccRCC in Japanese patients because all diabetic patients in our cohort were diagnosed with ccRCC. In addition, different environmental factors would partially contribute to mutagenesis in ccRCC and PRCC because we did not observe a significant difference in pathogenic germline mutations registered in the database (Supplementary Data 24). Prior chemotherapy is a risk factor for TFE3-RCC[63], and one patient had experience with it, but its contribution could not be determined from mutational signature analysis.

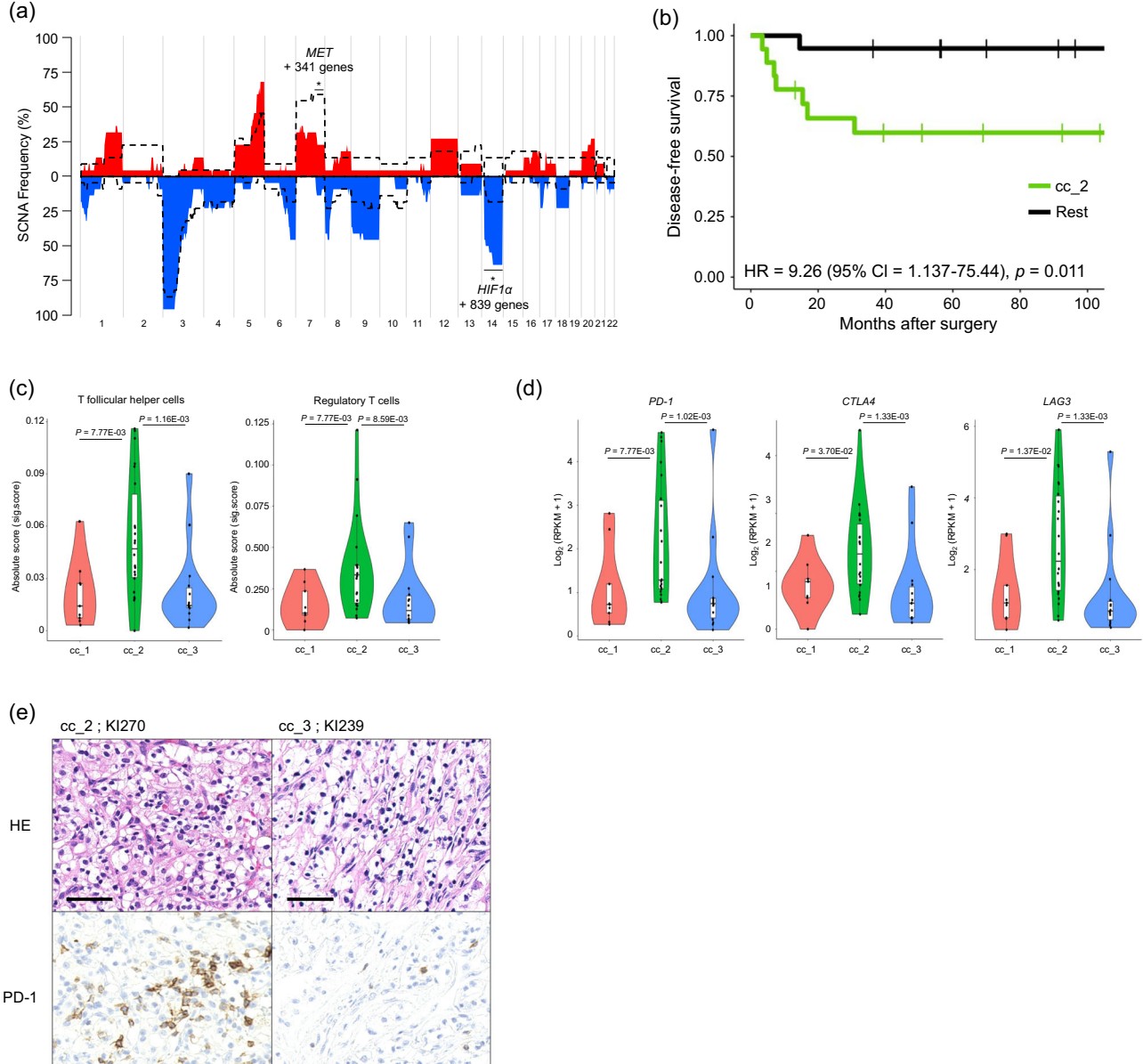

**Fig. 4 | Molecular features of ccRCC epi-subtypes. a** Frequencies of cc_2 cases and the remaining ccRCC cases with SCNAs. The histogram represents copy number gains and deletions in cc_2 in red and blue, respectively. The dotted lines show those in cc_1 and cc_3. *P*-values were calculated per cytoband using the two-sided Fisher's exact test. The areas where significant differences (*P*-value < 0.05) were observed in multiple cytobands consecutively were marked with *. *P*-values were shown in Supplementary Data 20. **b** Kaplan−Meier curves showing disease-free survival of cc_2 cases and the remaining cases. The *P*-value was calculated using the log-rank test. **c** CIBERSORTx absolute values of T follicular helper cells and regulatory T cells in ccRCC epi-subtypes. **d** Gene expressions of *PD-1*, *CTLA-4*, and *LAG3*. **e** Hematoxylin and eosin stain and immunohistochemistry (IHC) with PD-1 in sample IDs KI270 and KI239. Scale bars, 50 μm. IHC was performed for four samples, including all epi-subtypes, and similar results were shown. **c, b** Violin plots show the median (lines), IQR (boxes), ±1.5 × the IQR (whiskers), and individual samples (dots). *P*-values were calculated using the two-sided Wilcoxon rank-sum test. cc_1 *n* = 9, cc_2 *n* = 22, cc_3 *n* = 14. Source data are provided as a Source Data file.

Hierarchical clustering of ATAC-seq data revealed three epi-subtypes of ccRCC. This classification based on chromatin accessibility was informative because it reflected the immune environment and differed from the classifications based on transcriptome and methylation profiles reported in previous studies[16,64]. Tfh and Treg infiltration and the expression of immune checkpoint molecules such as *PD-1*, *CTLA-4*, and *LAG-3* were prominent in the cc_2 subtype. The degree of Tfh infiltration and expression levels of the above immune checkpoint molecules have been reported to be effective predictive markers of the response to immune checkpoint inhibitors and are consistent with a better prognosis in other solid cancers[65,66]. In contrast, in ccRCC, large-scale analyses show that Tfh and Treg infiltration and high expression of *PD-1* and *CTLA-4* are poor prognostic factors[13,67,68]. We demonstrated that cc_2 exhibited frequent loss of chr.14q, encoding *HIF1α*. Previous research has suggested that deletion of chr.14q is associated with a distinct immune environment, tumour heterogeneity, and poor prognosis of ccRCC[69–71], and this genomic aberration would influence these features via the abnormal chromatin accessibility seen in this study. Although a recent study reported that loss of chr.9p21.3 influenced the poor response to PD-1 blockade in ccRCC[21], in our study, this deletion was not statistically significant between cc_2 and the other two epi-subtypes. In cc_3, SWI/SNF family gene mutations and gain of

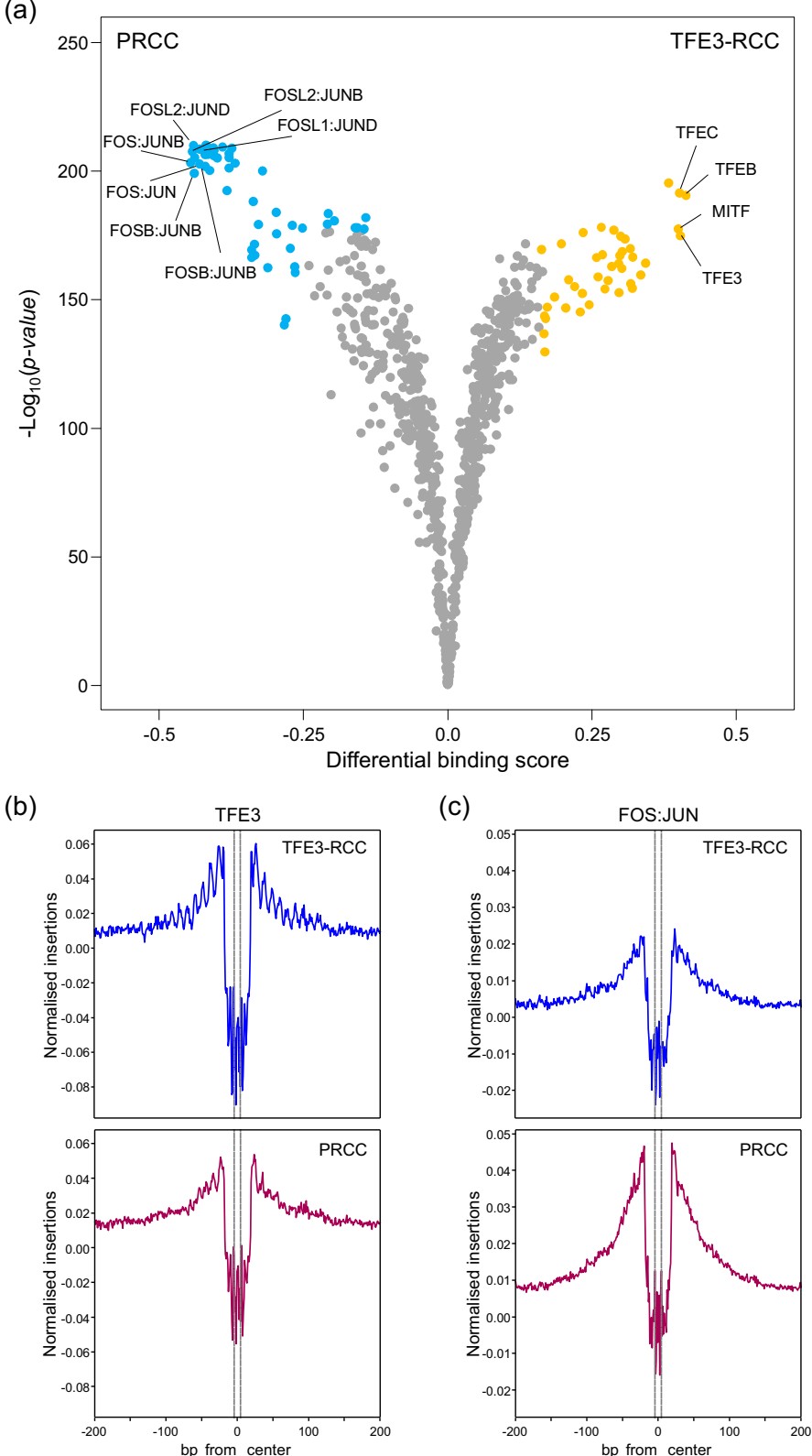

**Fig. 5 | Master transcriptional factors in PRCC and TFE3-RCC. a** Differential binding score comparing TFE3-RCC to PRCC. Each dot represents an individual motif sequence, and yellow and blue dots are predominant in TFE3-RCC and PRCC, respectively. One-sided *P*-values were calculated by TOBIAS. **b** TF footprint analysis of the TFE3 motif in TFE3-RCC and PRCC. **c** TF footprint analysis of the FOS:JUN motif in TFE3-RCC and PRCC. The region between the dotted lines shows the motif centre of FOS:JUN. Source data are provided as a Source Data file.

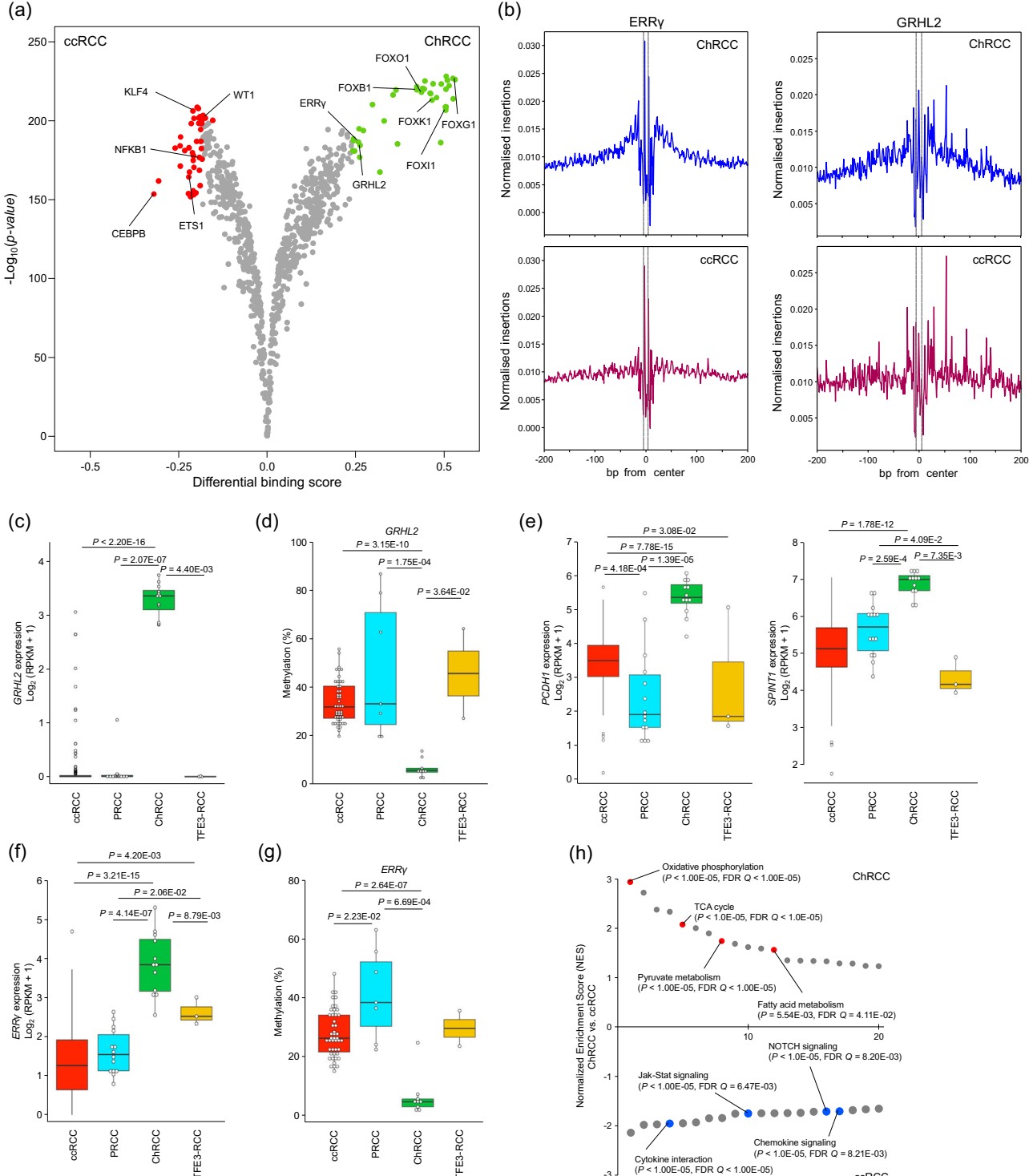

**Fig. 6 | Key molecules in ChRCC. a** Differential binding scores comparing ChRCC with ccRCC. Each dot represents an individual motif sequence, and green and red dots are predominant in ChRCC and ccRCC, respectively. One-sided *P*-values were calculated by TOBIAS. **b** In ChRCC and ccRCC, TF footprint analysis of the ERRγ and GRHL2 motifs, respectively. The regions between the dotted lines show the motif centres of ERRγ and GRHL2. **c** *GRHL2* gene expression in RCC. **d** Methylation status in the hypomethylated region (HMR), including the *GRHL2* promoter (chr.8: 102,503,600–102,507,618). **e** Gene expressions of *PCDH1* and *SPINT1* in RCC. **f** *ERRγ* gene expression in RCC. **g** The methylation status in HMR, including the *ERRγ* promoter (chr.1: 217,307,036–217,314,031). **h** Normalised enrichment scores (NES) were calculated by gene set enrichment analysis and the pathways registered with

the Kyoto Encyclopedia of Genes and Genomes (KEGG) for ChRCC and ccRCC. NES's positive and negative values indicate the enrichment of the gene set in ChRCC and ccRCC, respectively. **c, e, f** Box plots show the median (lines), IQR (boxes), ±1.5 × the IQR (whiskers), individual samples in PRCC, ChRCC, and TFE3-RCC (dots), and samples of the outliers in ccRCC (dots). *P*-values were calculated using the two-sided Wilcoxon rank-sum test. ccRCC *n* = 258, PRCC *n* = 14, ChRCC *n* = 12, TFE3-RCC *n* = 3. **d, g** Box plots show the median (lines), IQR (boxes), ±1.5 × the IQR (whiskers), and individual samples (dots). *P*-values were calculated using the two-sided Wilcoxon rank-sum test. ccRCC *n* = 46, PRCC *n* = 7, ChRCC *n* = 9, TFE3-RCC *n* = 2. Source data are provided as a Source Data file.

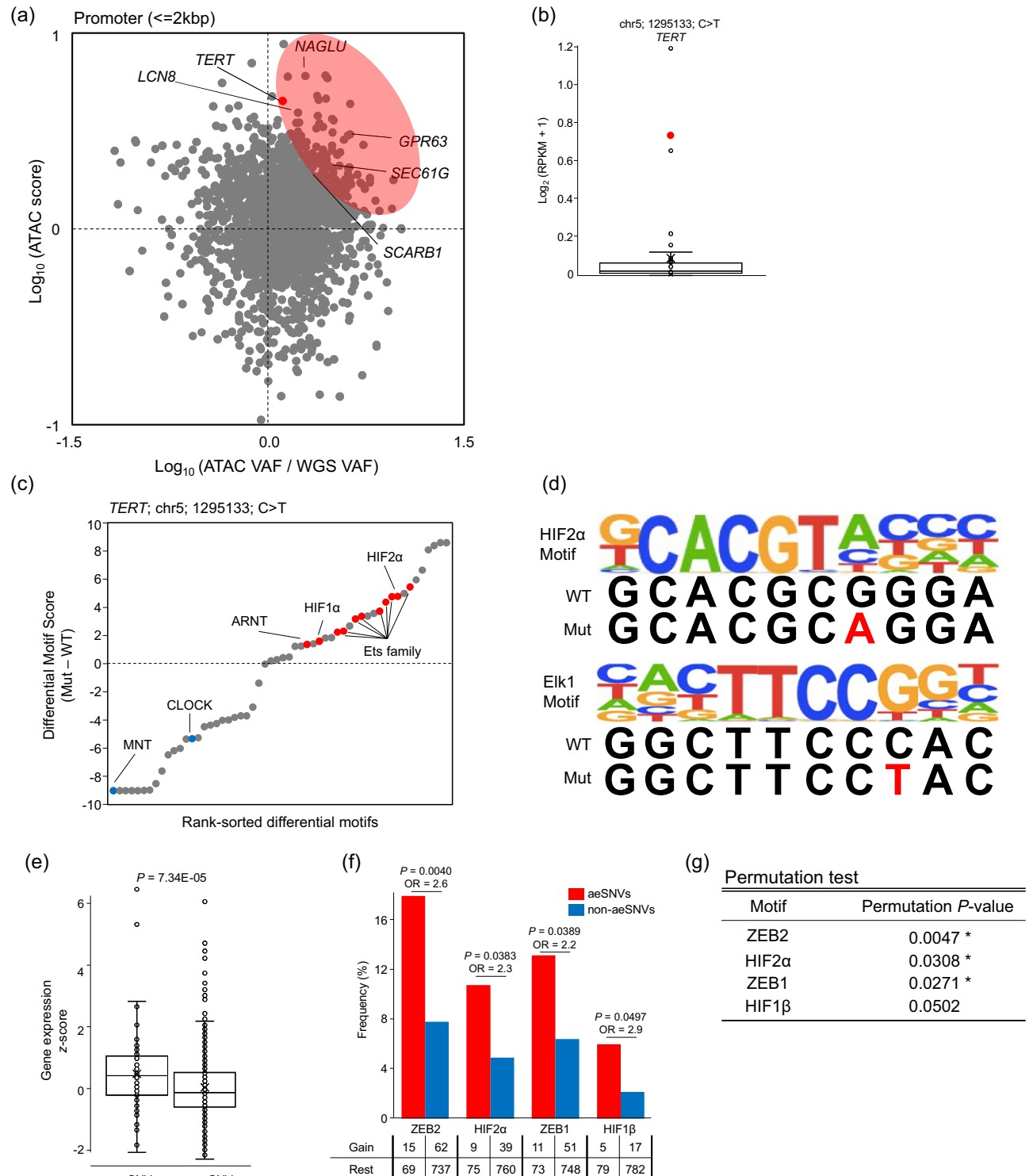

**Fig. 7 | Integrated analysis of genomic and epigenomic sequencing. a** The variant allele frequency (VAF) ratio of ATAC-seq to WGS and the degree of chromatin accessibility at the SNV's position. Each dot represents a somatic mutation in the promoter (between −2000 and +100 from TSS). The SNVs in the upper right quadrant are expected to increase chromatin accessibility. The SNVs of ccRCC in the red circle are defined as aeSNVs. **b** Gene expression of *TERT* in 47 ccRCC. Box plots show the median (lines), IQR (boxes), and ±1.5 × the IQR (whiskers), and each dot represents the expression of a single sample, and the red dot indicates the *TERT* promoter mutated sample. RPKM, reads per kilobase of exon model per million mapped reads. **c** Differential motif scores in the wild-type and mutant strings of the *TERT* promoter. Each dot represents an individual motif. The motif score is the degree of similarity between the sequence and the motif. **d** Sequence logos of

HIF2α and Elk1, showing both wild-type and mutant sequences. WT, wild-type sequence; Mut, mutant sequence. **e** Gene expression *z*-scores with aeSNVs and non-aeSNVs in ccRCC. Box plots show the median (lines), interquartile range (IQR; boxes), and ±1.5 × the IQR (whiskers). *P*-values were calculated using the two-sided Wilcoxon rank-sum test. aeSNV *n* = 83, non-aeSNV *n* = 758. **f** Events and frequencies of higher binding affinities by aeSNVs and non-aeSNVs. *P*-values and ORs were calculated using the two-sided Fisher's exact test. **g** Permutation test (10,000 replicates) of aeSNVs and non-ae-SNVs. One-sided permutation *P*-value, a frequency showing equal or greater significance than aeSNV. * indicate significant differences (permutation *P*-value < 0.05). Source data are provided as a Source Data file.

chr.7q were more frequent. Single-cell ATAC-seq data analysis showed that *PBRM1* and *BAP1* mutation changes chromatin accessibility in ccRCC[72]. In our study, the alterations of the SWI/SNF family genes, including *PBRM1*, were associated with enhancing chromatin accessibility of the *HIF* gene family and its downstream target genes (Fig. 3a), and this is consistent with the previous study. In addition, the contributions of HNF1A and HNF1B were more substantial in cc_3 than those of other epi-subtypes. This finding may indicate that the cell of origin in ccRCC is heterogeneous because HNF1 is an essential molecule for kidney development[73,74], or that this epi-subtype has undergone different driver alterations. The previous study using multi-platform-based clustering showed that the groups with unique immune environments and high expressions of hypoxia-related genes were detected as the same subtype[13]. However, in our analyses they were detected as separate subtypes.

The epigenomic profile of ChRCC was significantly different from those of ccRCC and PRCC. This is consistent with the hypothesis that the cell of origin in ChRCC is the distal convoluted tubule of nephron[9]. ChRCC had characteristics similar to those of intercalated cells in the distal nephron, including high expression of *FOXI1* and *KIT* in this study (Supplementary Fig. 11b). The high contribution of *GRHL2* is likely to reflect the cellular history of ChRCC, as well as *FOXI1* and *KIT*, as GRHL2 has been characterised as a distal nephron molecule[36–38]. In contrast, it is unclear whether ERRγ contributes more to cells in the distal nephron than in other parts[75,76]. In addition, ATAC-seq results from another study show that one of the ERR family genes is a characteristic molecule of ChRCC[77]. Therefore ERRγ may be acquired during tumorigenesis and progression rather than being an inherited property of ChRCC-derived cells. The candidates for the master transcriptional factor of ccRCC were Ets1 and HIF family genes (Fig. 6a), and these results were similar to the other ATAC-seq study. ATAC-seq results also suggested that AP-1 was highly involved in PRCC compared to ccRCC and TFE3-RCC (Fig. 5a and Supplementary Fig. 10a).

In addition to statistical analysis, functional annotation helps discover non-coding driver mutations[32,42,78]. We performed a combined analysis of WGS and ATAC-seq to identify the promoter SNVs that influence chromatin status. By comparing the two VAFs, the degree of increased chromatin accessibility caused by specific somatic mutations was evaluated. Although all chromatin accessibility-enhancing promoter mutations, including one *TERT* mutation, were singletons in this study, ZEB2 and HIF2α binding motifs overlapped significantly with the mutated sequences compared to the wild-type ones. Overexpression of HIF2α increases chromatin accessibility in the glycolysis-related genes in an in vitro study using leukaemic cells[79]. ZEB1/2 interacts with the Nucleosome Remodelling and Deacetylase complex, and regulates chromatin structure[80,81]. Our findings for HIF2α are consistent with a previous study, but as ZEB1/2 often acts suppressively, our results require further in-depth study. Accessibility-enhancing promoter mutations were enriched in nucleocytoplasmic transport genes registered in the GO term, and this feature was not observed for other promoter mutations. The lack of experimental validation is a limitation of this study, and whether these events drive ccRCC progression should be addressed in future studies.

Our study clarified the genomic and epigenomic profiles of the different RCC subtypes. These findings could be the basis for improved clinical and surgical treatment strategies for this disease.

## Methods

### Subjects and materials

We abided by the ethical standards of the institutional and national research committee and, with the 1964 Helsinki Declaration and its later amendments or comparable ethical standards. This study was approved by the Research Ethics Committee of the National Cancer Center (approval number: G2008-03), and written informed consent

to publish information that identifies individuals (including three or more indirect identifiers such as exact age, sex, medical centre the study participants attended, or rare diagnosis), collect the sample, analyze them, and deposit the sequencing data were obtained from all patients. The study included 287 patients diagnosed with ccRCC, PRCC, or ChRCC by pathological diagnosis. The information on gender and/or sex, number, and age of all participants in this study is presented in Supplementary Data 1. The specimens were stored in liquid nitrogen immediately after surgical resection. Moreover, the specimens for germline control were obtained from the adjacent normal kidney tissues.

### DNA extraction and WGS

Genomic DNA was extracted from tumours and matched normal tissues, and libraries with an insert size of 350 bp were prepared from 1 µg sonicated DNA using a TruSeq DNA PCR-free Kit (Illumina, San Diego, CA, USA). The libraries were sequenced on a NovaSeq 6000 instrument (Illumina) with paired-end reads of 150 bp. Sequencing reads were aligned to the human genome reference (GRCh37) using BWA-MEM v0.7.8[82] for both tumour and non-tumour samples. Using SAMtools v1.9 and in-house programmes, we removed potential PCR duplications and generated pileup files. Base selection cutoff values were set at a mapping quality score of ≥20 and a base quality score of ≥10.

### Mutation calling

We used an in-house developed method to reduce false positives. Somatic mutations were selected using two filtering conditions. The first condition was that the number of reads that detected a mutation in each tumour sample was ≥4, and at least one base quality score at the mutation position of these reads was >30. The second condition was that the VAF of the matched non-tumour samples was <0.03, with a read depth of at least eight. We then conducted a step to protect against the effects of sequence context-dependent errors. Sequence reads of all non-tumour samples were grouped to distinguish true positives from false positives. NVAF, defined as VAF in grouped non-tumour samples with a sequence depth ≥10 and VAF < 0.2, was calculated at each mutated genomic position.

The following filters were applied:

(i) NVAF < 0.03 in case of TVAF ≥ 0.15, or NVAF < 0.01 in case of 0.05 ≤ TVAF < 0.15

(ii) TVAF/NVAF ≥ 20

(iv) for non-tumour samples, 0.1 ≤ VAF < 0.002 at each mutated genomic position

At each mutated genomic position, the ratio of non-tumour samples with VAF ≥ 0.1 must be less than 0.002. Finally, mutations with a strand bias (between forward and reverse reads) >95% were discarded.

### Mutational signature analysis

The contribution of signal-mutational signatures was calculated using FitMS [https://signal.mutationalsignatures.com/analyse2] with GEL-kidney-SBS in the organ-specific signature[30]. Clustering cases based on mutational signature contribution were performed using unsupervised hierarchical clustering with cosine distance and Ward's linkage. The significance of the association between the signature contribution and clinical data was evaluated by multivariate analysis using the lm function in R v3.5.0.

### Copy number analysis

We used FACETS v0.6.0[83] to determine allele-specific copy numbers, purity, and ploidy, using the following parameters: snp.nbhd = 500, preProcSample's val = 50, and procSample's Cval = 300. Amplification was defined as genes containing segments with a copy number of ≥2 ×

ploidy. Gain was defined as genes containing segments with a copy number ≥1.5 × ploidy. Homozygous deletions were defined as genes with segments having a copy number of 0. Loss of heterozygosity was determined based on genes with segments with a minor allele copy number of 0. Finally, we defined the whole genome doubled sample as the case in which the proportion of the region with a significant allele copy number ≥2 was more than 50% of the whole genome, concerning a previous study[84]. We defined deletions in the whole genome duplicated samples as a copy number less than 2.0 × ploidy.

## RNA extraction and RNA-seq

RNA sequencing of 287 tumour samples (ccRCC, 258 cases; PRCC, 14 cases; ChRCC, 12 cases; TFE3-RCC, 3 cases) was performed using a HiSeq 2500 instrument (Illumina). Libraries were prepared using the NEBNext Ultra II Directional RNA Library Prep Kit (New England Bio-Labs, Ipswich, MA, USA). Paired-end reads were aligned using the Bowtie (v0.12.7) programme[85] with the −v 3 and -a options to allow mismatches ≤3 and to detect all multiple hits, respectively. After selecting the best hits with the correct distance and orientation, RPKM values were calculated.

## Detection of fusion genes

We used an in-house developed method to reduce false positives. For detecting fusion genes, 50-bp paired-end reads were preferred because their associated spacers were longer than those of 100-bp paired-end reads. Therefore, 100-bp paired-end reads were separated to generate 50-bp paired-end reads. Paired-end reads aligned uniquely to different genes with two or fewer mismatches were considered. Paired clusters indicating fusion transcripts were selected: Forward and reverse clusters, including paired-end reads, were constructed from end sequences aligned in each direction. Two reads were classified into the same cluster if their end positions were within 300 bp of each other. Clusters were discarded in which the leftmost read was more than 1000 bp from the rightmost read. Paired-end reads were selected if one end sequence fell within the forward cluster and the other end sequence fell within the reverse cluster (hereafter, these pairs of forward and reverse clusters are referred to as "paired clusters"). Paired-end reads classified into paired clusters were aligned to human reference RNA sequences using the BLASTN (v2.2.26) programme to select paired clusters, in which at least one pair of paired-end reads perfectly matched the human reference RNA sequences and to remove gene pairs that were mis-selected due to nucleotide variations. The cutoff of the expectation value was set at 1000 in BLASTN so that paired-end reads with low similarity to human reference RNA sequences could also be aligned. If one end sequence was aligned to a region of paired clusters and the other was aligned to the same RNA sequence with the correct spacing and orientation of the paired-end library, the gene pair was removed. Finally, the gene pairs with at least four paired-end reads were extracted.

## Immune-cell infiltration analysis

The CIBERSORTx web tool [https://cibersortx.stanford.edu] was used to assess tumour-infiltrating immune cells based on the transcripts per million (TPM) calculated from RNA-seq data. CIBERSORTx, with LM22 as the signature matrix, batch correlated B-mode option, LM22 source GEP as the optional source GEP, and 1000 permutations, was used to calculate immune cell infiltration scores of 22 cell types, including B cells, T cells, natural killer cells, macrophages, and dendritic cells.

## GSEA

The pathways registered in KEGG were tested using GSEA v4.1.0 to detect significant pathways. The number of permutations and types were set to 1000 and the gene_set, respectively. The cut-off q-value was set at 0.1.

## ATAC-seq

Using 72 tumour samples from our institute (ccRCC, 51 cases; PRCC, 9 cases; ChRCC, 10 cases; TFE3-RCC, 2 cases), ATAC-seq was performed on the Illumina HiSeq 2500 using 75-bp paired-end sequences. The libraries were prepared after nuclei extraction using iodixanol concerning a previous study[86], and amplified by PCR for 5–6 cycles using NEBNext High-Polymerase with Fidelity 2X PCR Master Mix (New England BioLabs). Primer data were shown in Supplementary Data 25.

One hundred and twenty-two cases were analysed, including 50 cases of RCC deposited in TCGA (ccRCC, 16 cases; PRCC, 34 cases). Sequencing reads were aligned to the human genome reference assembly hg19 using BWA-MEM v0.7.8[82]. TSS enrichment scores were calculated using the R package ChrAccR v0.9.11, and our six cases (ccRCC, four cases; PRCC, two cases) with TSS enrichment scores of less than five were excluded.

## Definitions of ATAC scores, differential motif scores, and aeSNVs

Of the 857,020 SNVs detected by WGS, SNVs annotated between −2000 and +100 bp from TSSs were targeted. The VAFs of each SNV position from the ATAC-seq were calculated. The ATAC score was calculated as an indicator of accessibility as follows:

ATAC score = sample (position reads/total reads)/mean histology (position reads/total reads).

To identify motifs whose binding affinity was predicted to be altered by the given SNVs, we constructed WT and Mut strings ±12 bp from each SNV, with lengths of 25 bp, using the hg19 BSGenome reference sequence. We then used FIMO in MEME Suite v5.0.5[87] and the known motifs in HOMER [http://homer.ucsd.edu/homer/] to calculate the motif positions and scores of both strings. Next, we extracted the sequences in WT and Mut that overlapped the SNVs with the highest motif scores and calculated the differential motif scores. Finally, we limited the motifs with the P-value < 0.01 in the Mut strings to remove motifs that would be expected not to bind to the region.

We extracted approximately 10% of the total SNVs of ccRCC as aeSNVs, as follows:

aeSNVs: $\log_{10}$ (ATAC VAF/WGS VAF) > 0.1, $\log_{10}$ (ATAC score) >0.1, and the top 125 (approximately 10% of total SNVs in ccRCC cases) ranked by $\log_{10}$ (ATAC VAF/WGS VAF) + (ATAC score).

We performed GSEA for genes with aeSNVs and non-aeSNVs using the R package clusterProfiler v.4.6.0, with the option pvalueCutoff = 0.05.

## Permutation test

We randomly extracted the 84 SNVs in ccRCCs SNVs using the sample function in R and calculated the number of SNVs with increased specific motif scores, such as ZEB2 and HIF2α. This test was repeated 10,000 times, and we counted the tests showing the number of SNVs with increased specific motif scores above or equal to aeSNVs. Finally, we calculated the permutation p-value, the number of these tests/10,000.

## Hierarchical clustering, motif analysis, and peak calling of ATAC-seq

All mapped reads were offset by +4 bp for the +strand and by −5 bp for the -strand to remove the bias of the Tn5 transposase cutting site. Using cancerPeaks[32], which consists of 501-bp tiling regions, a consensus set of ATAC-seq peaks identified in a large cancer dataset, we counted mapped reads using featureCounts[88] in Rsubread v2.4.3. Sequence count data were normalised using DESeq2 v1.31.14 and annotated using R packages TxDb.Hsapiens.UCSC.hg19.knownGene v3.2.2 and ChIPseeker v1.26.2[89] to obtain the distance from the TSS. We extracted the regions whose TSS distance was between −100,000 and +1000 bp and ranked the top 20,000 regions by row variance using

matrixStats v0.60.1 for clustering. findMotifsGenome.pl in HOMER v4.11 was used for motif analysis. Peak calling was performed using Macs2 v2.1.4[90] with the following options: –nomodel –shift –100 –extsize 200.

To validate our data, we performed footprint analysis using TOBIAS v0.12.10[33] with JASPAR 2022 vertebrate motifs [https://jaspar.genereg.net/downloads/], not HOMER motifs.

### EM-seq

EM-seq was conducted in 64 cases (ccRCC, 46 cases; PRCC, 7 cases; ChRCC, 9 cases; TFE3-RCC, 2 cases) that were used for ATAC-seq and passed quality control. The NEB Next Enzymatic Methyl-seq Kit (New England BioLabs) was used to generate libraries using 500 ng of DNA. Control libraries were then prepared by mixing pU19 DNA with methylated CpG and bacteriophage λ DNA with unmethylated CpG. Finally, all libraries were sequenced on a NovaSeq 6000 instrument (Illumina) using 150-bp paired-end sequences.

Sequencing reads were aligned to hg19 using Bismark v0.20.0[91] with the Bowtie option. We set the quality control cutoff that bacteriophage λ DNA converted more than 99.5% of unmethylated cytosine, and pU19 DNA methylated more than 96%, and all samples passed.

### HMR calling

Base-level methylation calling was performed using Bismark v0.20.0[91] with default parameters, as recommended in the Bismark User Guide for the library kit. Bases with C > T or G > A mutations based on the WGS results were excluded from the analysis on a per-sample basis. Unmethylated and low-methylated regions (UMR/LMR) were identified using MethylSeekR v1.22.0[92] with default parameters. HMR was defined as contiguous UMR/LMR in ≥3% of RCC samples identified by MethylSeekR, according to a previous study[93]. We limited the HMRs to ≥50 bp in this analysis.

### Immunohistochemistry (IHC)

Immunohistochemistry (IHC) for PD-1 was performed for each ccRCC epi-subtype. De-paraffinized sections were autoclaved in target retrieval solution pH9.0 for antigen retrieval. IHC analysis was performed automatically using AutostainerLink 48 (Agilent Technologies, Santa Clara, CA, USA) with the primary antibody PD-1 (catalogue number ab234444, colon NAT105; Abcam, Cambridge, UK) mouse monoclonal antibody at a dilution of 1:100. The antibody was validated by immunohistochemistry with the human lymph follicle.

### Reporting summary

Further information on research design is available in the Nature Portfolio Reporting Summary linked to this article.

## Data availability

Japanese 287 RCC cases' raw sequencing data (WGS 128, RNA-seq 287, ATAC-seq 72, Methyl-seq 64) generated in this study have been deposited in the European Genome-phenome Archive (EGA) under accession EGAS00001006919. These data are available under restricted access because they are personally identifiable data defined by Japan's Personal Information Protection Law. Requests for academic purposes only will be processed by the ICGC Data Access Compliance Office [https://docs.icgc-argo.org/docs/data-access/daco/applying] within ten business days. After access has been granted, the data is available for two years. TCGA data we used are deposited under dbGaP Study Accession phs000178. These data are available under restricted, and access to the data requires user certification through dbGaP-authorised access. PCAWG data are deposited in Synapse under accession syn11726616. These data are available under restricted and can be downloaded by the users registering in Synapse. Human genome reference (GRCh37) was downloaded from the UCSC genome browser [https://hgdownload.soe.ucsc.edu/downloads.html#human]. Source data are provided with this paper.

## Code availability

We used publicly available software for the analyses. All software used in this study is described in the Methods section and the accompanying Reporting Summary.

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

## Acknowledgements

We thank all patients who participated in this study. We also thank Y. Kobayashi and S. Shiba for clinical information collection and pathological advice, F. Hosoda, H. Chikuta, E. Arakawa, A. Hara, and A. Matsumoto for expert technical assistance, and Y. Sato and S. Ogawa for the advice on data analysis. We performed computations on the SHIROKANE supercomputer at the Institute of Medical Science, The University of Tokyo. This work was supported by grants from the Practical Research Project for Innovative Cancer Control from the Japan Agency for Medical Research and Development (AMED) (21ck0106547h000 to T.S.), National Cancer Center Research and Development Fund (2020-A-7 and 2023-A-5 to T.S.).

## Author contributions

T.S. designed the study. A.F., A.M. and Y.M. contributed to sample acquisition and pathological evaluation. A.F. and Y.A. managed library preparation and sequencing. A.F., N.H., Y.T., H.N. and M.S.A. analysed data. A.F., H.N., Y.T., H.N., Y.A., M.S.A., S.Y., T.U. and T.S. interpreted data and results. A.F., N.H., Y.T., H.N. and T.S. wrote the manuscript. All authors critically reviewed the manuscript.

## Competing interests

The authors declare no competing interests.
