## [Peer Review File · Nature Communications]

REVIEWER COMMENTS

Reviewer #1 (Remarks to the Author):

The manuscript by Fukagawa et al. performed multi-omics profiling and integrative analysis of different renal cell carcinoma (RCC) histological types of a cohort of 128 Japanese patients. Most of the findings recapitulate the previously published genomic and epigenomic features in RCC subtypes. Notably, the authors performed an integrative analysis of the paired genomic and epigenomic data to define accessibility enhancing SNVs (aeSNVs), which haven't been studied in RCC field, however limited validation has been performed to show the significance of this novel finding. Overall, we think the novel parts of the manuscript need to be consolidated with additional validations in the context of the published studies, therefore we would like the authors to address the following comments.

Major comments

1. The epigenomic analysis results of different RCC histological types is similar to this paper (<https://doi.org/10.1038/s41467-023-35833-5>). The authors only discussed very briefly about the results from this paper. A more comprehensive comparison of the results should be done in the discussion (what has been shown previously, what is supported and what is different).
2. The aeSNVs which is the novel part of the paper should be validated with experiments (establishing isogenic lines with or without aeSNVs to recapitulate the analysis findings), and/or with additional datasets.
3. It will be more convincing to include a cross-comparison of the scATAC-seq data of the corresponding cell origins in human kidney (e.g. <https://doi.org/10.1038/s41467-022-34255-z> and <https://doi.org/10.1038/s41586-023-05769-3>). This might help to understand if there are tumour specific epigenetic mis-regulation in different RCC types, rather than a reflection of the cellular history of the epigenetic landscape.
4. There is not a clear association of the epi-subtypes with driver mutations or SCNAs . Whether the observed epi-subtypes could be correlate with the TME features (e.g. could cc_2 subtype be just explained by more immune infiltration?), or there is a RCC cancer cell intrinsic mechanism contributing to the differences. More comprehensive work might be expanded from this point. The recent publication (<https://doi.org/10.1038/s41467-023-38547-w>) "Epigenetic and transcriptomic characterization reveals progression markers and essential pathways in clear cell renal cell carcinoma", which utilised snATAC-seq might be a good validation dataset for some of the findings regarding ccRCC epi-subtypes.

Minor comments

1. ZFH3 (along with SETD2 and TP53) is associated with advanced-stage cases, could the authors show if this is also the cases in other RCC genomic studies?
2. TOPORS is on Chr9p which is frequently lost in high-grade ccRCC. Could the authors clarify if the mutation is co-occurring with a Chr9p loss event, which leads to a homozygous inactivation of the gene?
3. The authors have used custom somatic variant calling instead of employing any existing benchmarked somatic variant calling algorithm like Mutect. Similarly, they have used in-house methodology for fusion genes detection. It would be necessary to explain in the methods if there is there any particular reason for this.
4. Whether the aeSNVs are clonal or subclonal? It will be useful to include the purity estimate and VAFs from WGS and ATAC-seq for the 125 aeSNVs?

Reviewer #2 (Remarks to the Author):

The authors have adequately responded to all my comments.

Reviewer #3 (Remarks to the Author):

My previous comments revolved around the fact that TCGA datasets were previously used to explore genomic and epigenomic integrative subtypes of renal cell carcinoma. In response to my comments, the authors have added some references to the previous TCGA studies, e.g., in the intro or discussion. No additional analyses or comparisons were carried out.

My issues with this manuscript begin with the title: "Genomic and epigenomic integrative subtypes of renal cell carcinoma." A more appropriate title would arguably be "Genomic and epigenomic integrative subtypes of renal cell carcinoma IN A JAPANESE COHORT." The study should build upon the previous findings reported from TCGA, e.g., to see how the Japanese cohort would reinforce the previous findings in (predominantly Caucasian/U.S.) TCGA cases or would be unique to the Japanese cases.

The authors report that mutational signatures differed among the histological types. Mutational signatures should be evaluated in TCGA. The PCAWG study by Stratton and colleagues (PMID: 32025018) evaluated signatures based on exomes across all TCGA patients, with results available via synapse

(<https://www.synapse.org/#!Synapse:syn11726616>). Results involve all three histologic types of RCC. Are the mutational signature patterns in both TCGA and Japanese cohorts comparable?

The ATAC-seq analysis involved a combination of TCGA and Japanese samples. Are the findings shared by both cohorts? In particular, regarding the three epi-subtypes of clear cell RCC, do these correspond to the molecular subtypes of ccRCC found in TCGA (PMID: 26947078). The first TCGA pan-RCC study found three multiplatform-based subtypes of ccRCC. The three ATAC-seq based subtypes should be overlapped with the TCGA-based subtypes, where we would expect a strong correspondence between the respective subtypes.

"We conducted an integrative analysis of ATAC-seq and WGS data to identify promoter mutations that affect chromatin accessibility." Did this analysis include TCGA samples? It is not clear from the write-up how many patients are involved in this analysis (we would assume it involves the Japanese cases). TCGA has WGS data for around 150 RCC patients (50 from each subtype). These data were analyzed as part of PCAWG, though data should also be available in the Genome Data Commons. Some of the other reviewer comments wanted to see a validation of some of the SNV findings. Analysis of the TCGA samples might provide this, although the overlap between TCGA WGS RCC and TCGA ATAC-seq RCC might be small.

Reviewer #1 (Remarks to the Author):

The manuscript by Fukagawa et al. performed multi-omics profiling and integrative analysis of different renal
cell carcinoma (RCC) histological types of a cohort of 128 Japanese patients. Most of the findings recapitulate
the previously published genomic and epigenomic features in RCC subtypes. Notably, the authors performed
an integrative analysis of the paired genomic and epigenomic data to define accessibility enhancing SNVs
(aeSNVs), which haven't been studied in RCC field, however limited validation has been performed to show
the significance of this novel finding. Overall, we think the novel parts of the manuscript need to be consolidated
with additional validations in the context of the published studies, therefore we would like the authors to address
the following comments.

Thank you for this important comment. We agree with the reviewer that the validation emphasizes the
novelty and uniqueness of our results. Regarding the experimental validation, to verify the change in
transcriptional activity, we need to compare all mutations that increase the binding motif score with the
random mutations (as a negative control) in renal cell carcinoma cell lines, which is beyond the scope
of the current analysis. Instead, we performed a permutation test to validate our results (Figure 7g), and
several transcription factors remained significant. This suggests that not all of these mutations are non-random
ones. Although these results provide some insight into the non-coding cancer mutations, we agree that they are
still not robust, so we have moved them to Figure 7 and avoided definitive wording in the text.

In this study, clear cell renal cell carcinoma could be classified into three epi-subtypes by ATAC-seq
results, and one epi-subtype was characterized by a unique immune environment. In this part, we have
performed immunohistochemistry for PD-1 as experimental validation (Figure 4e and Supplementary Figure
9d). In the other part, we have cited published studies and discussed similarities and differences with our results,
following the reviewer's comment below.

Major comments

1. The epigenomic analysis results of different RCC histological types is similar to this paper
(<https://doi.org/10.1038/s41467-023-35833-5>). The authors only discussed very briefly about the results from
this paper. A more comprehensive comparison of the results should be done in the discussion (what has been
shown previously, what is supported and what is different).

Thank you for the important comments. The study by Nassar et al. reported master transcription factors
in each histological subtype, which are partly similar to our results. In particular, the study showed that FKH2,
ESRRB, USF2, and TFCP2 are characteristic of chromophobe renal cell carcinoma (ChRCC). We have also
found that ESRR family genes are master transcription factors of ChRCC (Figure 3a and Figure 6a). As the
review previously suggested, we have cited this work and added the reproducibility of our results in lines 332-
336.

In addition, we detected Ets1 and HIF family genes as the key molecules of clear cell renal cell
carcinoma (ccRCC) (Figure 6a), and this result was also similar to that paper. Interestingly, papillary renal cell
carcinoma featured AP-1 compared to ccRCC and TFE3-translocated renal cell carcinoma (Figure 5a and
Supplementary Figure 10a), and this finding is different from that paper. We have added and discussed these
interesting findings in the main text (lines 336-339).

2. The aeSNVs which is the novel part of the paper should be validated with experiments (establishing isogenic
lines with or without aeSNVs to recapitulate the analysis findings), and/or with additional datasets.

Thank you for this important comment. We agree with the reviewer that the experimental validation
emphasizes the significance of our results. However, to verify the change in transcriptional activity, we need to
compare all mutations that increase the binding motif score with the random mutations (as a negative control)
in renal cell carcinoma cell lines, which is beyond the scope of the current analysis.

3. It will be more convincing to include a cross-comparison of the scATAC-seq data of the corresponding cell
origins in human kidney (e.g. <https://doi.org/10.1038/s41467-022-34255-z> and [https://doi.org/10.1038/s41586-](https://doi.org/10.1038/s41586-023-05769-3)
[023-05769-3](https://doi.org/10.1038/s41586-023-05769-3)). This might help to understand if there are tumour specific epigenetic mis-regulation in different

RCC types, rather than a reflection of the cellular history of the epigenetic landscape.

Thank you for the insightful comment. We agree with the reviewer, and have compared our results
with the scATAC-seq data from normal kidney. We found that the ERR family is not characteristic
transcriptional factors of distal nephron cells, and speculate that ERR γ may be acquired during tumorigenesis.
We have cited the paper entitled “Single cell transcriptional and chromatin accessibility profiling redefine
cellular heterogeneity in the adult human kidney”¹ and discussed our results in the revised manuscript (lines
332-336), following the previous reviewer’s comment.

The URL (<https://doi.org/10.1038/s41467-022-34255-z>) provided by the reviewer links to a paper
entitled “Defining cellular complexity in human autosomal dominant polycystic kidney disease by multimodal
single cell analysis.” This study focuses on the comparison between autosomal dominant polycystic kidney
disease (ADPKD) and normal kidney. The URL (<https://doi.org/10.1038/s41586-023-05769-3>) links to a study,
“An atlas of healthy and injured cell states and niches in the human kidney.” This study compared between
healthy and injured cells in the human kidney using multi-omics analyses. As mentioned above, we have already
cited the excellent study entitled “Single cell transcriptional and chromatin accessibility profiling redefine
cellular heterogeneity in the adult human kidney”¹, which showed the characteristics of each of the cells that
make up the nephrons. This study deepened our discussion and would be more appropriate to cite in terms of
its purpose than the two studies. In addition, the same normal kidney tissue samples were analyzed in "Single
cell transcriptional and chromatin accessibility profiling redefine cellular heterogeneity in the adult human
kidney" (which we cited) and "Defining cellular complexity in human autosomal dominant polycystic kidney
disease by multimodal single cell analysis." Therefore, we have not cited them additionally in this part.

4. There is not a clear association of the epi-subtypes with driver mutations or SCNAs . Whether the observed
epi-subtypes could be correlate with the TME features (e.g. could cc 2 subtype be just explained by more
immune infiltration?), or there is a RCC cancer cell intrinsic mechanism contributing to the differences. More
comprehensive work might be expanded from this point. The recent publication

<https://doi.org/10.1038/s41467-023-38547-w> “Epigenetic and transcriptomic characterization reveals
progression markers and essential pathways in clear cell renal cell carcinoma”, which utilised snATAC-seq
might be a good validation dataset for some of the findings regarding ccRCC epi-subtypes.

Thank you for the important comment. To verify the association between epi-subtypes and driver
mutations/ somatic copy number alterations (SCNAs), we performed multivariate analysis by comparing one
epi-subtype with the other. The result showed that several nonsynonymous mutations (SWI/SNF family genes)
and SCNAs (chromosome 7q amplification) correlated with the epi-subtype (Supplementary Table 16 and lines
156-163). The result suggests that the characteristics of RCC cancer cells contribute to the difference in the
immune environment.

We performed CIBERSORTx using TPM and the appropriate option instead of RPKM, as suggested
by the reviewer. The results showed that immune cell infiltration was not significantly different among the three
epi-subtypes, and infiltration of T follicular helper cells and regulatory T cells was higher in cc_2 than in the
other epi-subtypes (Supplementary Figure 9b).

The study by Wu et al. shows that *PBRM1* and *BAP1* mutation alter chromatin accessibility in ccRCC.
In our study, the alterations of the SWI/SNF family genes, including *PBRM1*, were associated with increased
chromatin accessibility of the *HIF* gene family and its downstream target genes, such as *VEGFA* and *NDRG1*
(Figure 3a). Our results are consistent with the previous study, and we have cited that paper and discussed our
results in the revised manuscript (lines 316-320).

The URL (<https://doi.org/10.1038/s41467-023-38547-w>) provided by the reviewer links to a paper
entitled “Topology of vibrational modes predicts plastic events in glasses”. This paper had very little overlap
with our study and was not mentioned in the main text.

Minor comments

1. ZFH3 (along with SETD2 and TP53) is associated with advanced-stage cases, could the authors show if
this is also the cases in other RCC genomic studies?

Thank you for your comment. Previous studies (D'Avella, C et al. Urol Oncol 2020, Liu, L. et al.

Biosci Trends. 2017) have also reported that *TP53* and *SETD2* are involved in the acquisition of aggressive
features^{2,3}, but *ZFH3* has not been reported in previous renal cancer genomic studies.

2. TOPORS is on Chr9p which is frequently lost in high-grade ccRCC. Could the authors clarify if the mutation
is co-occurring with a Chr9p loss event, which leads to a homozygous inactivation of the gene?

Thank you for this important comment. We have checked the mutation data and confirmed that two
out of the four TOPORS mutated cases had the loss of chr.9p. We have added this data in Figure 1a.

3. The authors have used custom somatic variant calling instead of employing any existing benchmarked
somatic variant calling algorithm like Mutect. Similarly, they have used in-house methodology for fusion genes
detection. It would be necessary to explain in the methods if there is there any particular reason for this.

Thank you for your comment. We have used the in-house developed somatic variant calling and fusion
gene detection to reduce false positives, and these methods have been used in our previous publications (Totoki,
*Y. et al., Nat Genet* 2023, Yachida S et al. *Cancer Discov.* 2022)^{4,5} (line 395 and line 438). The algorithms are
also described in the Methods section.

4. Whether the aeSNVs are clonal or subclonal? It will be useful to include the purity estimate and VAFs from
WGS and ATAC-seq for the 125 aeSNVs?

Thank you very much for this interesting comment. We estimated the tumor purity and calculated the
adjusted variant allele frequencies of aeSNVs (Supplementary Table 22). The subclonal (aVAF<40%) aeSNVs
were 46%, and there is no specificity in the clonality.

Reviewer #2 (Remarks to the Author):

The authors have adequately responded to all my comments.

Thank you for your comment. Your constructive feedback has helped us to have a better discussion in

this manuscript. We appreciate your kind review.

Reviewer #3 (Remarks to the Author):

My previous comments revolved around the fact that TCGA datasets were previously used to explore genomic
and epigenomic integrative subtypes of renal cell carcinoma. In response to my comments, the authors have
added some references to the previous TCGA studies, e.g., in the intro or discussion. No additional analyses or
comparisons were carried out.

We appreciate the comments. Following the reviewer's suggestion, we have cited previous studies
from The Cancer Genome Atlas (TCGA) and added discussions of the similarities and differences. We agree
with the reviewer that the additional analyses for the TCGA dataset increase the novelty of our results. We have
performed additional analyses to validate our data using the Pan-Cancer Analysis of Whole Genomes (PCAWG)
samples, as suggested by the reviewer (Supplementary Figure 2a and Supplementary Table 14).

My issues with this manuscript begin with the title: "Genomic and epigenomic integrative subtypes of renal cell
carcinoma." A more appropriate title would arguably be "Genomic and epigenomic integrative subtypes of renal
cell carcinoma IN A JAPANESE COHORT." The study should build upon the previous findings reported from
TCGA, e.g., to see how the Japanese cohort would reinforce the previous findings in (predominantly
Caucasian/U.S.) TCGA cases or would be unique to the Japanese cases.

We appreciate the important comments. We agree with the reviewers that the title "Genomic and
epigenomic integrative subtypes of renal cell carcinoma" is not appropriate because of the insufficient
comparison between Japanese renal cell carcinoma (RCC) and cases from other countries. Therefore, we have
changed the title to "Genomic and epigenomic integrative subtypes of Japanese renal cell carcinoma."
Similarities with and differences from previous large-scale studies are discussed in each main text section
(lines 280-283, lines 316-320, and lines 323-326).

The authors report that mutational signatures differed among the histological types. Mutational signatures
should be evaluated in TCGA. The PCAWG study by Stratton and colleagues (PMID: 32025018) evaluated
signatures based on exomes across all TCGA patients, with results available via synapse

<https://www.synapse.org/#!Synapse:syn11726616>). Results involve all three histologic types of RCC. Are the
mutational signature patterns in both TCGA and Japanese cohorts comparable?

Thank you for the important comment. We agree with the reviewer that the addition of the TCGA data
set emphasizes the significance of our results in terms of transethnic research. We performed mutational
signature analysis for PCAWG RCC samples (Supplementary Figure 2a and Supplementary Table 14). Similar
to our cohort, SBS107 and SBS125 were predominant in ccRCC, and SBS125 contributed more to PRCC
compared to ccRCC, but SBS107 contributed less. Furthermore, the contribution of SBS125 was limited in
ChRCC. In contrast, SBS117 and SBS141 had high contributions in Japanese samples and were not detected in
PCAWG cases. These results suggest that these signatures are characteristic of the Japanese cases. We have
added these findings to the main text (lines 127-132 and 280-283) and cited the PCAWG study by Stratton and
colleagues (PMID: 32025018).

The ATAC-seq analysis involved a combination of TCGA and Japanese samples. Are the findings shared by
both cohorts? In particular, regarding the three epi-subtypes of clear cell RCC, do these correspond to the
molecular subtypes of ccRCC found in TCGA (PMID: 26947078). The first TCGA pan-RCC study found three
multiplatform-based subtypes of ccRCC. The three ATAC-seq based subtypes should be overlapped with the
TCGA-based subtypes, where we would expect a strong correspondence between the respective subtypes.

We appreciate the insightful comment. In the ATAC-seq analysis (Figure 3a), we analyzed Japanese
cases and 50 cases (ccRCC 16 cases, PRCC 34 cases) deposited at TCGA. As a result, ccRCC could be classified
into three epi-subtypes. However, as the reviewer pointed out, it remains to be verified whether the same
findings are observed in genomic aberrations and gene expression in the TCGA data set.

In this TCGA study (PMID: 26947078), ccRCC was divided into three subtypes using the multi-
platform method (DNA methylation, DNA copy alteration, mRNA expression, miRNA expression, and protein
expression), and some of the characteristics of the subtypes were similar to those in this study. For example, a
group with high expression of hypoxia-related genes and epithelial-mesenchymal transition-related genes was
detected, which are very similar to cc_3 in this study (Figure 3a and Supplementary Figure 8). The genetic

aberrations associated with this subtype were loss of chr.9 in the TCGA study. In contrast, t nonsynonymous
mutations of the SWI/SNF complex were associated in the present study (Supplementary Table 16). In addition,
the expression of checkpoint-related genes (e.g., PD-1, CTLA4) differed from that in the present study. In our
study, the expression of hypoxia-related genes was not high in the epi-subtype with a characteristic immune
environment. These differences could be due to the sequencing technologies (ATAC-seq) used for clustering.
We have described these similarities and differences in the main text (lines 323-326).

"We conducted an integrative analysis of ATAC-seq and WGS data to identify promoter mutations that affect
chromatin accessibility." Did this analysis include TCGA samples? It is not clear from the write-up how many
patients are involved in this analysis (we would assume it involves the Japanese cases). TCGA has WGS data
for around 150 RCC patients (50 from each subtype). These data were analyzed as part of PCAWG, though data
should also be available in the Genome Data Commons. Some of the other reviewer comments wanted to see a
validation of some of the SNV findings. Analysis of the TCGA samples might provide this, although the overlap
between TCGA WGS RCC and TCGA ATAC-seq RCC might be small.

Thank you for the important comment. We apologize for the misleading wording. As the reviewer
pointed out, the analysis was performed for our cases with overlapping WGS and ATAC-seq data. We have
clearly stated this in the manuscript (line 219).

The ATAC-seq data of ccRCC deposited at TCGA are 16 cases, and all 16 cases did not have WGS
data. Therefore, we could not validate our data using the TCGA data set. Instead, following the comments of
the other reviewer, we performed a permutation test to validate our results (Figure 7g), and several transcription
factors remained significant.

References

- 1. Muto, Y. *et al.* Single cell transcriptional and chromatin accessibility profiling redefine cellular
heterogeneity in the adult human kidney. *Nat Commun* **12**, 2190 (2021).
- 2. D'Avella, C., Abbosh, P., Pal, S.K. & Geynisman, D.M. Mutations in renal cell carcinoma. *Urol Oncol*
**38**, 763-773 (2020).
- 3. Liu, L. *et al.* Loss of SETD2, but not H3K36me3, correlates with aggressive clinicopathological
features of clear cell renal cell carcinoma patients. *Biosci Trends* **11**, 214-220 (2017).
- 4. Totoki, Y. *et al.* Multiancestry genomic and transcriptomic analysis of gastric cancer. *Nat Genet* **55**,
581-594 (2023).
- 5. Yachida, S. *et al.* Comprehensive Genomic Profiling of Neuroendocrine Carcinomas of the
Gastrointestinal System. *Cancer Discov* **12**, 692-711 (2022).

REVIEWERS' COMMENTS

Reviewer #1 (Remarks to the Author):

The authors stated that validation aeSNVs is beyond the scope of the paper but they could mention this as a limitation of their study in the discussion.

The title of the manuscript is now changed to “Genomic and epigenomic integrative subtypes of Japanese renal cell carcinoma”. It would be appropriate to change it to “Genomic and epigenomic integrative subtypes of renal cell carcinoma in a Japanese cohort”.

Reviewer #3 (Remarks to the Author):

The authors have adequately addressed most of my comments. The part that I think the authors could do a bit more to address regards the three epi-subtypes of clear cell RCC and how these subtypes would relate to TCGA results. TCGA had found three subtypes of ccRCC. In the first TCGA marker of ccRCC [PMID: 23792563], ccRCC was stratified into better, worse, and intermediate patient survival based on molecular patterns. In the pan-RCC study led by TCGA investigators [PMID: 26947078], molecular subtyping using combined omics platforms stratified ccRCC into three subtypes, corresponding largely to the patient survival groups from the 1st marker paper. It's very likely that the three ATAC-seq based ccRCC subtypes in the current study, based on both TCGA and Japanese samples, correspond to the three ccRCC subtypes. In the manuscript revision, the authors note some similarities in some key molecular associations that would parallel the respective classifications but try to leave it to future studies to formally demonstrate robust associations. Actually, the authors would be able to do this now using the data in hand, as described below.

1. There are 16 TCGA ccRCC tumors with ATAC-seq data. These 16 tumors were classified by both ATAC-seq and the TCGA pan-RCC paper. It's a relatively small number, but it is worth it to show a confusion matrix comparing the ATAC-seq subtype assignments with the pan-RCC subtype assignments. [Look up "Confusion matrices with more than two categories" at https://en.wikipedia.org/wiki/Confusion_matrix]
2. Across all 50 TCGA RCC cases with ATAC-seq data, six clusters (methyl-clusters 1-6) of methylation status were identified by the authors. The confusion matrix suggested above could be expanded to represent all 50 cases and all histologic types. (Likely, PRCC subtypes would overlap between pan-RCC multi-omic and ATAC-seq).
3. For the 66 Japanese ccRCC cases with ATAC-seq data, the authors should be able to classify these by the pan-RCC multi-omic subtype, using the top differential mRNAs in the supplementary data from the pan-RCC study. The cases with ATAC-seq presumably have RNA-seq data as well. Then, make a confusion

matrix between the previous ATAC-seq subtypes and the subtypes as classified using the pan-RCC mRNA classifier.

4. For Figure 3a, including a sample color bar for TCGA versus Japanese case would be helpful (like the color bar representing histologic type).

Reviewer #1 (Remarks to the Author)

The authors stated that validation aeSNVs is beyond the scope of the paper but they could mention this
as a limitation of their study in the discussion.

Thank you for your comment. We agree with the reviewer that the study of accessibility-
enhancing SNVs lacks and needs experimental validation. We have clearly stated the need for
experimental validation in future studies in the main text (line 356).

The title of the manuscript is now changed to “Genomic and epigenomic integrative subtypes of
Japanese renal cell carcinoma”. It would be appropriate to change it to “Genomic and epigenomic
integrative subtypes of renal cell carcinoma in a Japanese cohort”.

Thank you for your comment. As suggested by the reviewer, we have changed the title to
"Genomic and epigenomic integrative subtypes of renal cell carcinoma in a Japanese cohort".

Reviewer #3 (Remarks to the Author)

The authors have adequately addressed most of my comments. The part that I think the authors could
do a bit more to address regards the three epi-subtypes of clear cell RCC and how these subtypes would
relate to TCGA results. TCGA had found three subtypes of ccRCC. In the first TCGA marker of ccRCC
[PMID: 23792563], ccRCC was stratified into better, worse, and intermediate patient survival based
on molecular patterns. In the pan-RCC study led by TCGA investigators [PMID: 26947078], molecular
subtyping using combined omics platforms stratified ccRCC into three subtypes, corresponding largely
to the patient survival groups from the 1st marker paper. It's very likely that the three ATAC-seq based
ccRCC subtypes in the current study, based on both TCGA and Japanese samples, correspond to the
three ccRCC subtypes. In the manuscript revision, the authors note some similarities in some key
molecular associations that would parallel the respective classifications but try to leave it to future
studies to formally demonstrate robust associations. Actually, the authors would be able to do this now
using the data in hand, as described below.

**We appreciate the insightful comment. As suggested by the reviewer, we performed additional**
**analyses to assess the association between the two classifiers. We describe the additional analyses and**
**results below.**

1. There are 16 TCGA ccRCC tumors with ATAC-seq data. These 16 tumors were classified by both
ATAC-seq and the TCGA pan-RCC paper. It's a relatively small number, but it is worth it to show a
confusion matrix comparing the ATAC-seq subtype assignments with the pan-RCC subtype
assignments. [Look up "Confusion matrices with more than two categories"
at https://en.wikipedia.org/wiki/Confusion_matrix]

**Thank you for your comment. We created the matrix to assess the association between our epi**
**subtypes and pan-RCC subtypes (Figure A in the response figure file). Two of the P-e.2 cases also**
**showed PRCC-like features in our classification by chromatin structure. However, the association**

between the two classifiers is not strong.

2. Across all 50 TCGA RCC cases with ATAC-seq data, six clusters (methyl-clusters 1-6) of
methylation status were identified by the authors. The confusion matrix suggested above could be
expanded to represent all 50 cases and all histologic types. (Likely, PRCC subtypes would overlap
between pan-RCC multi-omic and ATAC-seq).

We appreciate the comment and apologize for the misunderstanding. We analyzed DNA
methylation profiles of 64 cases in our cohort, but did not include the TCGA samples (lines 143-146).
Methyl clusters 1-6 have been identified in Japanese RCC cases. As our analysis method (whole-
genome enzymatic methyl sequencing) is different from that of the TCGA data (methylation array),
we could not analyze the two datasets simultaneously.

3. For the 66 Japanese ccRCC cases with ATAC-seq data, the authors should be able to classify these
by the pan-RCC multi-omic subtype, using the top differential mRNAs in the supplementary data from
the pan-RCC study. The cases with ATAC-seq presumably have RNA-seq data as well. Then, make a
confusion matrix between the previous ATAC-seq subtypes and the subtypes as classified using the
pan-RCC mRNA classifier.

Thank you for the comment. As the reviewer suggested, we performed a clustering analysis
on our 287 samples using the pan-RCC mRNA classifier (Figure B in the attached file). We were able
to classify Japanese ccRCC into three groups (TCGA_subtype 1-3). We found that cc_1 in our study
was enriched in TCGA_subtype_3 (Figure D). cc_2 was more frequently observed in both
TCGA_subtype_1 and TCGA_subtype_2. We also found that about 400 genes used in this TCGA
classifier showed extremely low diversity (yellow in Figure B) in our Japanese samples. Although we
could not definitively conclude whether the pan-RCC mRNA classifier was optimized for the TCGA
set and not adequate for the Japanese cases, we believe that these results would not be very informative.

4. For Figure 3a, including a sample color bar for TCGA versus Japanese case would be helpful (like
 the color bar representing histologic type).

Thank you for your comment. We agree with the reviewer that the annotation bar would show
 the similarities and differences across transethnic groups. We added a color bar to identify the source
 of samples easily (Figure 3a).

Figure Legends

(A) Matrix indicating where the 16 TCGA ccRCC cases were classified into epi-subtype of this study and subtype in the TCGA study. The number represents the number of cases classified into the subtypes. (B) Heatmap of mRNA across all RCC samples in this study using TCGA mRNA classifier's gene. (C) Matrix indicating where the 45 Japanese ccRCC cases were classified into epi-subtype of this study and subtype of Figure B. The number represents the number of cases classified into the subtypes. (D) Table indicating the results of two-sided Fisher's exact test for TCGA subtypes and epi-subtypes.